EMBO
Molecular Medicine

# CIB2, defective in isolated deafness, is key for auditory hair cell mechanotransduction and survival

Vincent Michel[1,2,3,†], Kevin T Booth[4,5,†], Pranav Patni[1,2,3,†], Matteo Cortese[1,2,3], Hela Azaiez[4], Amel Bahloul[1,2,3], Kimia Kahrizi[6], Ménélik Labbé[1,2,3], Alice Emptoz[1,2,3], Andrea Lelli[1,2,3], Julie Dégardin[3,7], Typhaine Dupont[1,2,3], Asadollah Aghaie[1,2,3,8], Danuta Oficjalska-Pham[1,2,3], Serge Picaud[3,7], Hossein Najmabadi[6], Richard J Smith[4], Michael R Bowl[9], Steven DM Brown[9], Paul Avan[10], Christine Petit[1,2,3,11] & Aziz El-Amraoui[1,2,3,*] (ID)

## Abstract

Defects of CIB2, calcium- and integrin-binding protein 2, have been reported to cause isolated deafness, DFNB48 and Usher syndrome type-IJ, characterized by congenital profound deafness, balance defects and blindness. We report here two new nonsense mutations (pGln12* and pTyr110*) in *CIB2* patients displaying nonsyndromic profound hearing loss, with no evidence of vestibular or retinal dysfunction. Also, the generated *CIB2*[−/−] mice display an early onset profound deafness and have normal balance and retinal functions. In these mice, the mechanoelectrical transduction currents are totally abolished in the auditory hair cells, whilst they remain unchanged in the vestibular hair cells. The hair bundle morphological abnormalities of *CIB2*[−/−] mice, unlike those of mice defective for the other five known USH1 proteins, begin only after birth and lead to regression of the stereocilia and rapid hair-cell death. This essential role of CIB2 in mechanotransduction and cell survival that, we show, is restricted to the cochlea, probably accounts for the presence in *CIB2*[−/−] mice and *CIB2* patients, unlike in Usher syndrome, of isolated hearing loss without balance and vision deficits.

**Keywords** CIB proteins; human and mouse deafness; Usher syndrome diagnosis
**Subject Categories** Genetics, Gene Therapy & Genetic Disease; Neuroscience

## Introduction

Biallelic mutations of *CIB2*, encoding calcium- and integrin-binding protein 2, have been reported to cause isolated deafness in DFNB48 and deaf-blindness in type 1J Usher syndrome (USH1J; Riazuddin *et al*, 2012). Seven disease-causing *CIB2* mutations have been reported to date, one of which, c.192G>C (Glu64Asp), has been reported in USH1J patients (Riazuddin *et al*, 2012; see Fig 1A). USH1 is the most severe form of the three clinical subtypes of USH (USH1-3) and is characterized by congenital profound deafness, bilateral vestibular dysfunction and retinitis pigmentosa of prepubertal onset leading to blindness (Bonnet & El-Amraoui, 2012). Ten USH genes have been identified (see http://hereditaryhearingloss.org/). They encode six USH1 proteins, myosin VIIa (USH1B), harmonin (USH1C), cadherin-23 (USH1D), protocadherin-15 (USH1F), sans (USH1G) and CIB2 (USH1J); three USH2 proteins, usherin (USH2A), Adgvr1 (USH2C) and whirlin (USH2D); and one USH3 protein, clarin-1 (USH3A) (Weil *et al*, 1995, 2003; Eudy *et al*, 1998; Bitner-Glindzicz *et al*, 2000; Verpy *et al*, 2000; Ahmed *et al*, 2001; Alagramam *et al*, 2001; Bolz *et al*, 2001; Bork *et al*, 2001; Adato *et al*,

1 Génétique et Physiologie de l'Audition, Institut Pasteur, Paris, France
2 Unité Mixte de Recherche- UMRS 1120, Institut National de la Santé et de la Recherche Médicale, Paris, France
3 Sorbonne Universités, UPMC Univ Paris06, Paris, France
4 Molecular Otolaryngology and Renal Research Laboratories, Department of Otolaryngology- Head and Neck Surgery, University of Iowa, Iowa City, Iowa
5 Department of Molecular Medicine, Carver College of Medicine, University of Iowa, Iowa City, Iowa
6 Genetics Research Center, University of Social Welfare and Rehabilitation Sciences, Tehran, Iran
7 Retinal information processing – Pharmacology and Pathology, Institut de la Vision, Paris, France
8 Syndrome de Usher et Autres Atteintes Rétino-Cochléaires, Institut de la Vision, Paris, France
9 Mammalian Genetics Unit, MRC Harwell Institute, Oxford, UK
10 Laboratoire de Biophysique Sensorielle, Faculté de Médecine, Biophysique Médicale, Centre Jean Perrin, Université d'Auvergne, Clermont-Ferrand, France
11 Collège de France, Paris, France
*Corresponding author. Tel: +33 145688892; E-mail: aziz.el-amraoui@pasteur.fr
†These authors contributed equally to this work

2002; Weston *et al*, 2004; Ebermann *et al*, 2007; Riazuddin *et al*, 2012).

CIB2 is one of four members of a family of calcium- and integrin-binding proteins (CIB1, CIB2, CIB3 and CIB4). All members of this family have elongation factor-hand (EF-hand) domains able to bind $Ca^{2+}$ ions, and their N-terminal regions bind to the α-chain of integrin heterodimers (Naik *et al*, 1997; Hager *et al*, 2008; Riazuddin *et al*, 2012; Leisner *et al*, 2016; Jacoszek *et al*, 2017). CIB2 contains

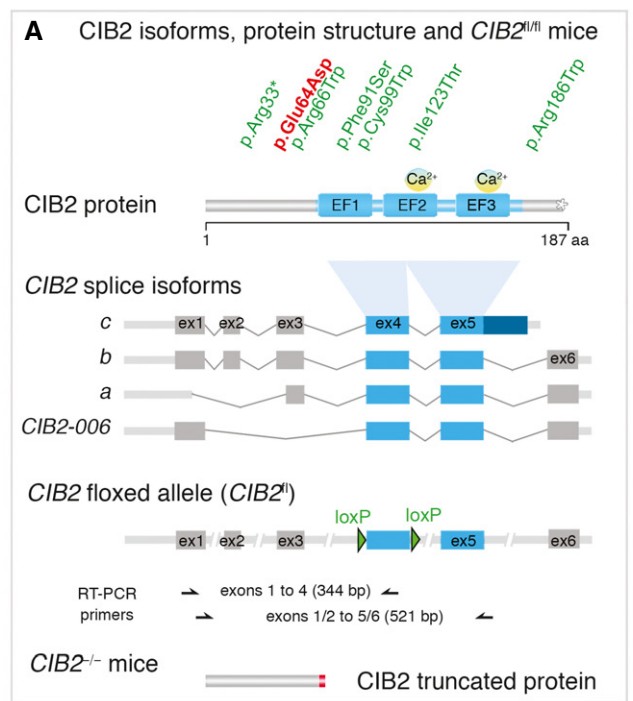

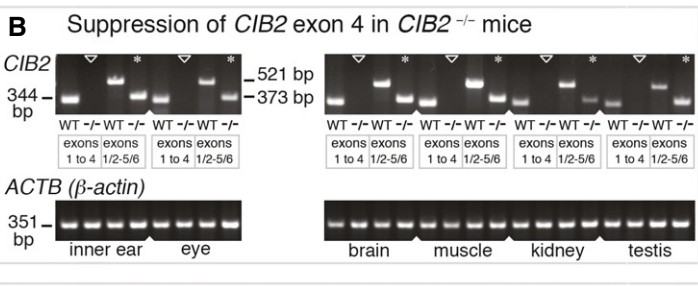

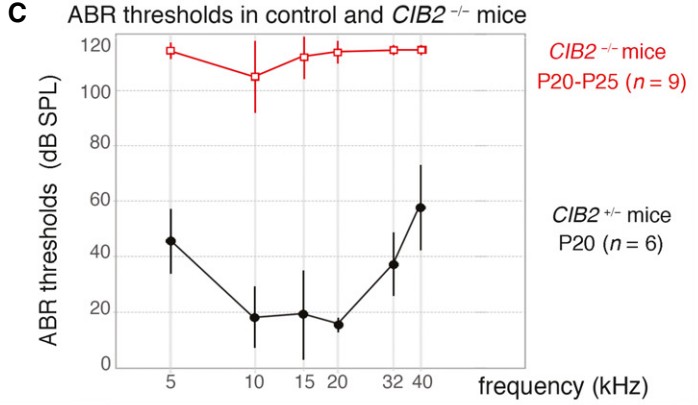

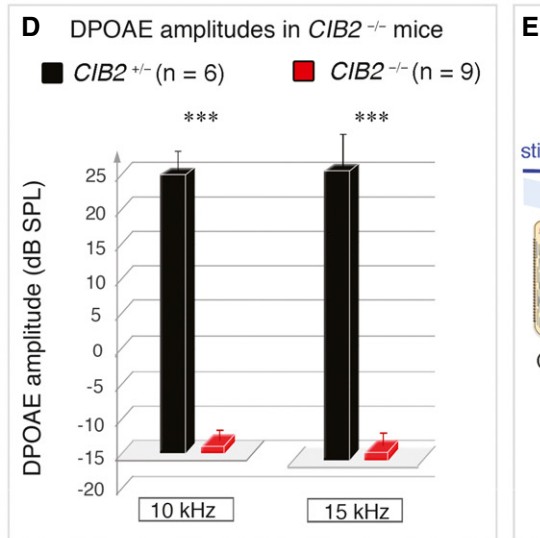

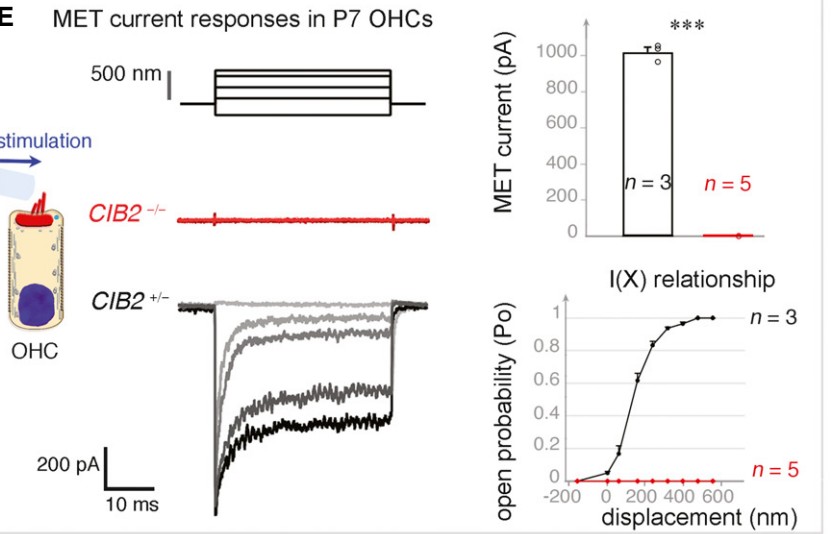

**Figure 1. CIB2 isoforms and hearing sensitivity in *CIB2*$^{-/-}$ mice.**

A    Domain structure of the CIB2 protein, indicating the positions of the *CIB2* mutations in USH1J (red) and DFNB48 (green) patients. The CIB2-floxed mice, *CIB2*$^{fl/fl}$, were engineered by adding LoxP sites on either side of exon 4, which is common to all four known *CIB2* transcripts.

B    RT–PCR analysis confirming the loss of *CIB2* exon 4-containing transcripts in the inner ear, eye, brain, muscle, kidney and testis of *CIB2*$^{-/-}$ mice. β-Actin was used as an endogenous control.

C, D    ABR thresholds (C) and DPOAE amplitudes (D) in *CIB2*$^{+/-}$ (dark, n = 6) and *CIB2*$^{-/-}$ (red, n = 9) P20-P25 mice. ABR thresholds in *CIB2*$^{-/-}$ mice exceeded 100 dB SPL (mean ± SD), indicating profound deafness. (D) DPOAE amplitudes were absent in *CIB2*$^{-/-}$ P20 mice at 10 and 15 kHz (red) (Mann–Whitney, ***P = 0.002 for both 10 and 15 kHz).

E    MET responses in OHCs from *CIB2*$^{+/-}$ and *CIB2*$^{-/-}$ P7 mice. The left panels show the mechanical stimulation protocol, with examples of MET currents for each genotype. In the right panels, the MET current values and mean amplitude-displacement relationships (I(X)) (mean ± SD) in *CIB2*$^{+/-}$ (black, n = 3 cells) and *CIB2*$^{-/-}$ (red, n = 5) mice highlight the absence of a MET response in *CIB2*$^{-/-}$ OHCs (Welch's unpaired *t*-test, ****P = 0.0007).

three EF-hand domains, only the second and third of which have been shown to bind $Ca^{2+}$ (Fig 1A). CIB2 has been shown to interact with myosin VIIa and whirlin (Riazuddin *et al*, 2012), an USH1 and USH2 protein, respectively, but the precise roles of this protein remain unknown.

Physiological, morphological and molecular analyses of mutant mice lacking USH1 proteins have shed light on the roles of USH1 proteins in the development and functioning of the hair bundle, the mechanoreceptive antenna by which the hair cells convert sound-evoked mechanical stimuli into electrical signals transmitted to the brain (Boeda *et al*, 2002; Siemens *et al*, 2002; Kazmierczak *et al*, 2007; Lefèvre *et al*, 2008; Grillet *et al*, 2009; Michalski *et al*, 2009; Bahloul *et al*, 2010; Caberlotto *et al*, 2011; Grati & Kachar, 2011; Pepermans *et al*, 2014). The hair bundle consists of 50–300 F-actin-filled stereocilia arranged in three rows that increase in height towards a true cilium (the kinocilium) and are held together by different types of interstereociliary fibrous links (Bonnet & El-Amraoui, 2012; Mathur & Yang, 2015; Richardson *et al*, 2011; see also Fig EV1A–C). Studies of hair cells have shown that USH1 proteins form molecular networks constituting hair bundle links, which they anchor to the actin filaments of the stereocilia, during both development and adulthood (Boeda *et al*, 2002; Siemens *et al*, 2002; Sollner *et al*, 2004; Adato *et al*, 2005; Lagziel *et al*, 2005; Michel *et al*, 2005; Kazmierczak *et al*, 2007; Lefèvre *et al*, 2008; Michalski *et al*, 2009; Bahloul *et al*, 2010; Caberlotto *et al*, 2011; Grati & Kachar, 2011; Pepermans *et al*, 2014). Cadherin-23 and protocadherin-15, in particular, form transient lateral links interconnecting the stereocilia together and also the stereocilia to the kinocilium in developing hair bundles (Boeda *et al*, 2002; Siemens *et al*, 2002; Adato *et al*, 2005; Lagziel *et al*, 2005; Michel *et al*, 2005; Lefèvre *et al*, 2008). Myosin VIIa, harmonin and sans have been shown to anchor these links to the actin filament core of the stereocilium (Boeda *et al*, 2002; Siemens *et al*, 2002; Weil *et al*, 2003; Lefèvre *et al*, 2008; Grillet *et al*, 2009; Michalski *et al*, 2009; Bahloul *et al*, 2010; Caberlotto *et al*, 2011; Grati & Kachar, 2011). In mature hair cells, USH1 proteins also play a crucial role in the mechanoelectrical transduction (MET) machinery, with protocadherin-15 and cadherin-23 forming the tip-link (Kazmierczak *et al*, 2007; Pepermans *et al*, 2014). This fibrous link runs from the tip of one stereocilium to the side of its taller neighbour and controls the opening probability of the MET channels located at the tips of the stereocilia (Pickles *et al*, 1984; Howard & Hudspeth, 1988; Pickles & Corey, 1992).

Owing to the hearing, balance and visual phenotypes that characterize USH1 patients (Boughman *et al*, 1983; Millan *et al*, 2011), we investigated the role of CIB2 in the inner ear and retina. To this purpose, we generated *CIB2* mutant mice (*CIB2*$^{-/-}$ mice) through an embryonic and ubiquitous suppression of exon 4, which is common to all CIB2 splice isoforms (see Fig 1A). We found that the absence of a functional CIB2 causes profound deafness without the vestibular and retinal phenotypes in both mouse and human. Molecular and morpho-functional analyses of the inner ear in *CIB2*$^{-/-}$ mice showed that, despite the normal shaping of the hair bundle and correct targeting of USH1 in the early developing hair bundle, no MET responses could be recorded in the cochlear outer hair cells. Unlike the embryonic disorganization of the hair bundle in other USH1 mutants, the lack of CIB2 leads to deleterious defects that

occur at postnatal stages and only in the hair cells of the hearing organ, but not the vestibular apparatus. Altogether, our findings demonstrate that CIB2 variants may not lead to Usher syndrome, suggesting that caution should be taken when providing genetic counselling for patients with *CIB2* mutations.

# Results

## *CIB2*$^{-/-}$ mice are profoundly deaf

The *CIB2* gene has four to six exons and encodes four main splice isoforms, a, b, c and CIB2-006 (Fig 1A). We generated *CIB2*$^{-/-}$ mice, using *CIB2*$^{fl/fl}$ mice with two flox sites flanking exon 4, which is common to all *CIB2* isoforms and predicted to encode the EF-hand domain region (see Materials and Methods and Fig 1A). These mice were crossed with *PGK-Cre* mice expressing Cre under the control of the early and ubiquitously active phosphoglycerate kinase-1 (PGK) gene promoter (Lallemand *et al*, 1998), to obtain an early constitutive ablation of *CIB2* exon 4 (Fig 1B), predicted to produce a truncated protein (aa 1–65) lacking all three EF domains (see Fig 1A). Similar results were obtained in functional and structural analyses of *CIB2*$^{fl/fl}$, *CIB2*$^{+/-}$ and *CIB2*$^{+/+}$ mice, so these mice were used indiscriminately as controls.

We first tested hearing function in *CIB2*$^{-/-}$ P20 mice by measuring auditory brainstem responses (ABR) to pure tones of frequencies between 5 and 40 kHz (Fig 1C and D). In *CIB2*$^{-/-}$ mice, ABR thresholds in response to tone bursts increased markedly over the 5- to 40-kHz frequency range to a sound pressure level (SPL) of over 100 dB, versus 20–40 dB SPL in age-matched control mice (Fig 1C). We also assessed the distortion product otoacoustic emissions (DPOAEs) elicited by two-tone stimuli. These DPOAEs, which probe the activity of outer hair cells (OHCs), the mechanical amplifiers of the sound stimulus, were detected in age-matched control mice, but were absent in *CIB2*$^{-/-}$ P20 mice (Fig 1D). Consistent with the lack of DPOAEs in these mutant mice, the cochlear microphonic potential (CM) response, which reflects the sound-induced transducer potentials of OHCs in the cochlear base region and is used as an indicator of OHC MET at mature stages (Patuzzi *et al*, 1989; Cheatham *et al*, 2011), was also found to be absent. We further investigated OHC activity at earlier stages, by mechanically stimulating the hair bundles of P7 OHCs in the apical third of the cochlea and then measuring MET currents by the whole-cell patch-clamp technique (see Materials and Methods). In control OHCs, the mean peak amplitude of the MET current was about $1018.6 \pm 27.2$ (mean $\pm$ SD; $n = 3$), whereas in *CIB2*$^{-/-}$ OHCs, no MET current was recorded ($n = 5$) ($P < 0.0001$; Welch's *t*-test) (Fig 1E).

Together, these findings demonstrate that the absence of CIB2 totally disrupts the MET activity of the auditory hair cells, leading to early onset total hearing loss.

## CIB2 is present in hair bundle stereocilia and at the apical surface of cochlear hair cells

As *CIB2*$^{-/-}$ mice were profoundly deaf, we investigated the precise subcellular distribution of CIB2, in the inner ear. Using a commercial anti-CIB2 antibody (ab111908, Abcam), we found that the pattern of immunostaining was similar for cochlear hair cells from

wild-type mice and mutant mice, demonstrating a lack of specificity of this antibody. We therefore generated a polyclonal antibody directed against a CIB2-specific region in the 2nd EF-hand domain (aa 129–153, accession number NP_006374.1; see Materials and Methods). This antibody labelled the stereocilia and the apical surface of hair cells in wild type, but not $CIB2^{-/-}$ mice, consistent with specific labelling in the sensory hair cells (Fig 2A–C). At P7, immunostaining of the stereocilia was concentrated mainly at the basal body of the kinocilium and in the hair bundle stereocilia

(Fig 2A). Analyses at later stages showed that CIB2 was located in the tip region of the stereocilia and at the apical surface of hair cells around the cuticular plate, as shown by immunostaining in P10 and P20 wild-type, but not in $CIB2^{-/-}$ mice (Fig 2B and C). Similar analyses of the vestibular organs revealed an absence of significant CIB2 immunostaining in the hair bundles of the vestibular hair cells (Fig 2D). Together, these findings indicate that CIB2 is a cochlear hair-cell protein located mainly in the stereocilia and at the apical surface of hair cells.

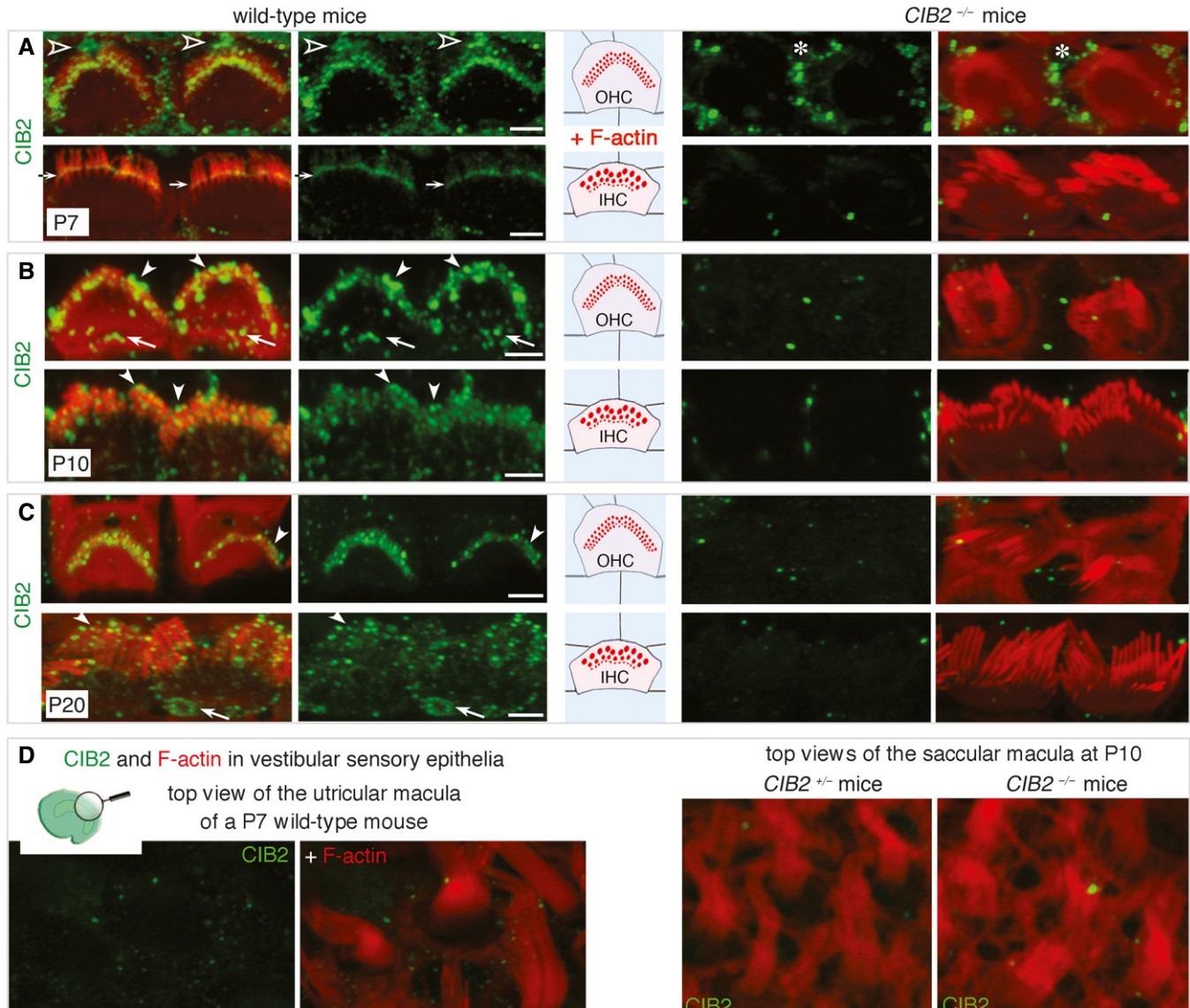

**Figure 2.   CIB2 in the cochlear hair bundles and the apical surface of hair cells.**

A–C   Location of CIB2 in F-actin-labelled (red) IHCs and OHCs. (A) On P7, specific labelling (absent from $CIB2^{-/-}$ mice, right panels) was observed in the basal body of the kinocilium (open arrowheads) at the apical surface of hair cells, and in the stereocilia, mostly in the basolateral region (arrows). Some unspecific labelling is observed in the supporting cells (asterisk in A). (B, C) On P10 (B) and P20 (C), CIB2 immunostaining was detected in different regions of the stereocilia, including their tips (arrowheads), and at the apical surface of hair cells (arrows), around the cuticular plate.

D   Regardless of the stage analysed (P7, P10 or P20), no significant immunostaining was observed in the vestibular hair cells.

Data information: Scale bars: 2 μm.

## CIB2$^{-/-}$ mice display progressive postnatal base-to-apex degeneration of cochlear hair bundles

Previous studies on mutant mice lacking USH1 proteins—myosin VIIa, harmonin, cadherin-23, protocadherin-15 and sans—revealed a fragmentation of hair bundles, initiated at embryonic stages and accounting for the profound deafness and balance phenotypes (Lefèvre *et al*, 2008; see also Fig 3A). We therefore investigated possible hair bundle defects underlying the deafness of CIB2$^{-/-}$ mice by scanning electron microscopy from birth onwards. At birth, the auditory sensory organ of CIB2$^{-/-}$ mice displayed no defect in the patterning of the sensory epithelium into well-organized three rows of OHCs and one row of IHCs (Fig 3B and C). On P3, the CIB2$^{-/-}$ cochlear hair bundles had a cohesive appearance similar to that of the hair cells of control mice, consistent with the presence of lateral interstereociliary links in these bundles (Fig 3D and E). Throughout the cochlea, the kinocilium was positioned normally at the vertex of the hair bundle in all auditory hair cells, with the hair bundles of CIB2$^{-/-}$ mice displaying the typical 3–4 rows of stereocilia arranged in a "staircase pattern" as in wild-type mice (see Fig 3C and E). The preservation of this normal organization, throughout the cochlear IHCs and OHCs of CIB2$^{-/-}$ mice, at least until P6 (Fig 3F), contrasts with the severe fragmentation of the auditory hair bundle over the same period typical of USH1$^{-/-}$ mice lacking any of the five main USH1 proteins, as shown here for an absence of the USH1G protein sans (Fig 3A and G; see also Lefèvre *et al*, 2008).

The first notable morphological defect in CIB2$^{-/-}$ mice, observed at the P6-P7 transition period in the auditory hair cells of CIB2$^{-/-}$ mice, was the presence at the base of the cochlea of rounded "horse-shoe"-shape bundles lacking their typical V-shape observed in age-matched controls (Fig 3F). The staircase pattern of the three rows of stereocilia was still preserved on P7, but the stereocilia in the shortest row in OHC bundles were heterogeneous in length (Fig 3F). At this stage, the IHC hair bundles at the cochlear base had an abnormal wavy shape, but, unlike OHCs, all the stereocilia within the same row were of the same length (Fig 3F). On the following days, more marked changes in hair bundle shapes rapidly occurred in both OHCs and IHCs, extending from the base to the apex of the cochlea (Figs 4A and B, and EV2A and B). On P9, most OHCs had discontinuous horseshoe-like shaped hair bundles, due to the loss of the centrally located stereocilia at the vertex of the bundle (Fig EV2B). On P18, the short row stereocilia had almost entirely disappeared in both IHC and OHC hair bundles, whereas those in the middle row were much shorter than usual, with some missing entirely (Figs 4A and B, and EV2B). CIB2$^{-/-}$ IHC hair bundles were not split like those of the OHCs, but they displayed misaligned stereocilia rows, and the stereocilia within rows had heterogeneous lengths. Also, many IHC bundles at this terminal mature stage still retained their kinocilia, whereas the wild-type IHCs lost this structure at post-hearing onset (beyond P14) stages (Figs 4C and D, and EV2C). On P90 and P120, only sporadic fused stereocilia or residual knoblike protrusions were observed on some of the remaining IHC stereociliary bundles of the mid-basal cochlea (Figs 5A and B, and EV3A–D). We used the scanning electron microscopy micrographs to quantify the number of IHC and OHC stereocilia bundles present at the mid-basal region of the cochlea in CIB2$^{+/-}$ and CIB2$^{-/-}$ P120 mice. As shown in Fig 5C, there was a near-complete loss of IHC

and OHC bundles on P120 (Fig 5C). The CIB1 protein has been implicated in cell survival (Hattori *et al*, 2009; Kawarazaki *et al*, 2014). The degeneration of the hair bundles in CIB2$^{-/-}$ mice thus prompted us to check for apoptosis in the cochlea. We used the TUNEL assay in P20 CIB2$^{+/-}$ and CIB2$^{-/-}$ mice. Unlike the hair cells in CIB2$^{+/-}$ mice (Fig 5D), dozens of hair cells displayed positive TUNEL staining in the cochlea of CIB2$^{-/-}$ mice (Fig 5E).

Together, our findings clearly show that, unexpectedly for an USH1 protein, CIB2 is dispensable for the early cohesion and shaping of the developing hair bundle. At mature stages, a functional CIB2 is required for the maintenance of hair-bundle function and structure and for hair-cell survival.

## Unlike the USH1 proteins, CIB2 is dispensable for vestibular functions in mouse

The presence of vestibular areflexia and the prepubertal onset of retinitis pigmentosa leading to blindness are defining clinical features of USH1 syndrome. Indeed, USH1 patients typically start walking later than their peers, generally between 18 and 24 months of age, and older patients frequently experience accidental injuries or have difficulty with activities requiring balance (Boughman *et al*, 1983; Petit, 2001; Bonnet & El-Amraoui, 2012; Mathur & Yang, 2015). We investigated whether, as reported for other USH1 proteins, the lack of CIB2 also led to vestibular dysfunction. We first explored MET at P7 in vestibular hair cells, comparing the results with those described above for OHCs in CIB2$^{-/-}$ and control mice (Fig 6A and B). The MET current responses of CIB2$^{-/-}$ utricular hair cells (UHCs) were indistinguishable from those of control UHCs (Fig 6A). The mean peak amplitude MET current was 328 ± 60 pA (mean ± SD; $n = 5$), in mutant UHCs, versus 320 ± 32 pA in control UHCs ($n = 5$; $P = 0.39$, Welch's $t$-test) (Fig 6B). We further investigated the physiological role of CIB2 in the vestibular system, by subjecting CIB2$^{-/-}$ mice to systemic behaviour tests at the ages of 3, 7 and 11 months (Hardisty-Hughes *et al*, 2010). These tests included platform, suspension, contact righting and swimming tests. Regardless of the test used, no clear difference between the two genotypes was observed; for example, CIB2$^{-/-}$ mice never displayed the trunk curling posture typically found in other USH1 mice with impaired vestibular function (Fig 6C). We quantified the exploratory behaviour of CIB2$^{-/-}$ mice in an open-field test system, with Ethovision behavioural tracking software (Noldus Information Technology, Wageningen). As expected, the Ush1g$^{-/-}$ mice displayed typical hyperactivity and circling behaviour consistent with the vestibular defects observed in USH1 mice defective for myosin VIIa, harmonin, cadherin-23, protocadherin-15 and sans (Lefèvre *et al*, 2008). By contrast, the CIB2$^{-/-}$ mice displayed no circling behaviour, at 3, 7 or 11 months of age (see Fig 6D). Their displacement was similar to that of CIB2$^{+/-}$ mice (Fig 6D), providing strong evidence for a lack of vestibular deficit in these mice. Supporting this proposal, ultrastructural analyses showed that, by contrast to observations for the cochlea, there were no abnormalities of hair bundle organization in the vestibular epithelia, regardless of the stage analysed in CIB2$^{-/-}$ mice (Figs 6E and F, and EV3E and F).

Together, these results show that CIB2 is dispensable for correct vestibular hair-cell functioning and survival. The lack of CIB2 does not lead to any vestibular structural or behavioural abnormalities irrespective of the stages analysed, contrasting strongly with the

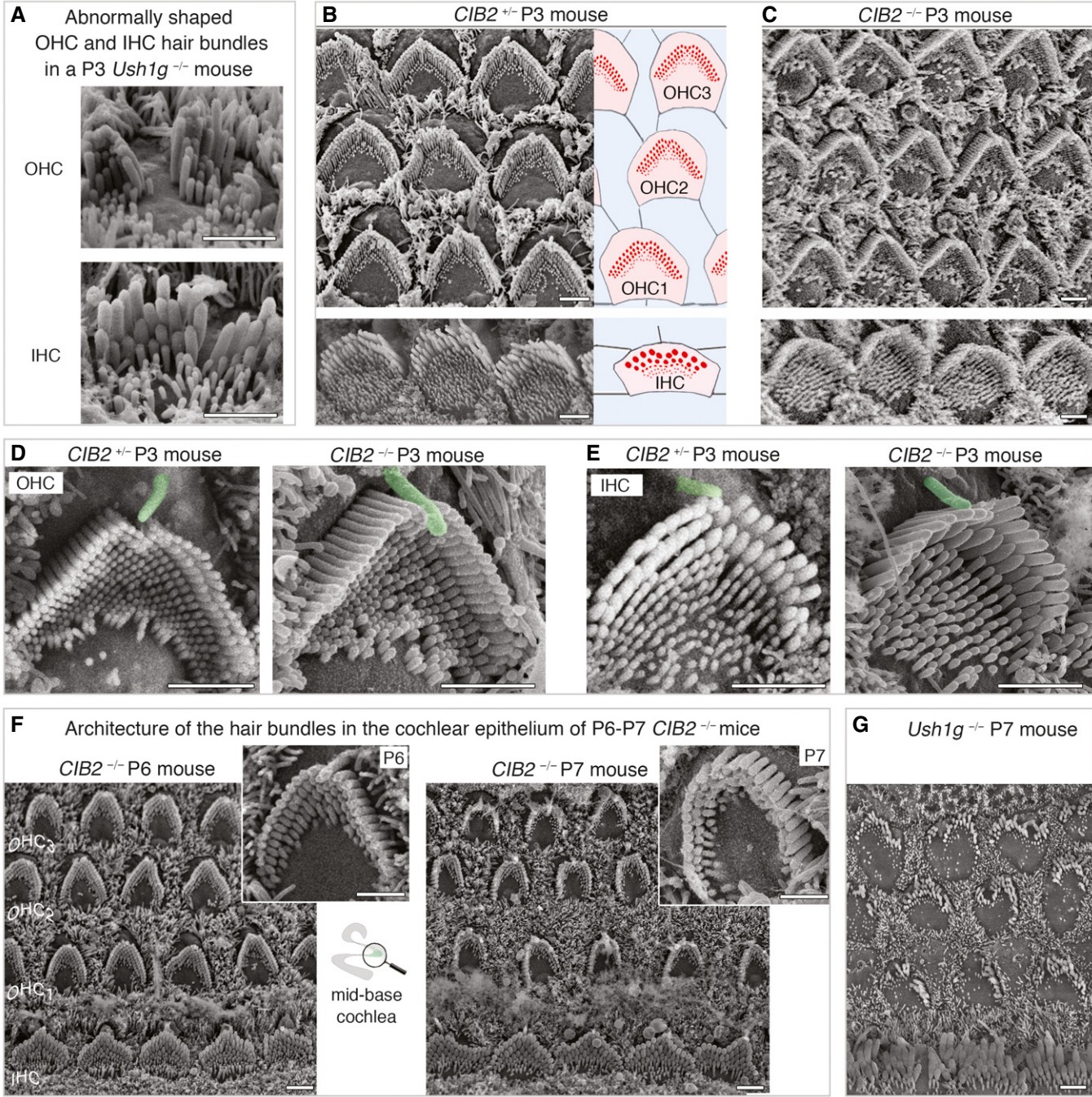

**Figure 3.  Architecture of cochlear hair bundles in control *CIB2⁺/⁻* and *CIB2⁻/⁻* mice.**

A    Scanning electron microscopy micrographs of misshapen IHC and OHC hair bundles from a P3 *Ush1g⁻/⁻* mouse.

B, C    Top views of the mid-base cochlear sensory epithelium from *CIB2⁺/⁻* (B) and *CIB2⁻/⁻* (C) P3 mice, showing the normal features of the developing hair bundles despite the absence of CIB2.

D, E    Representative hair bundles from OHCs and IHCs from the two genotypes show the normal position of the kinocilium, normal shape of the bundle, stereociliary cohesion and a typical staircase organization.

F    The first clear structural bundle abnormalities are observed from P6 onwards, with the typical V-shaped hair bundles of OHCs at the cochlear base on P6 (left panels) changing to a horseshoe-like shape on P7 (right panels). The *CIB2⁻/⁻* stereocilia of the short row are of different lengths in the deformed OHCs.

G    On P7, all *Ush1g*-defective IHCs and OHCs display abnormally shaped hair bundles.

Data information: Scale bars: 2 μm.

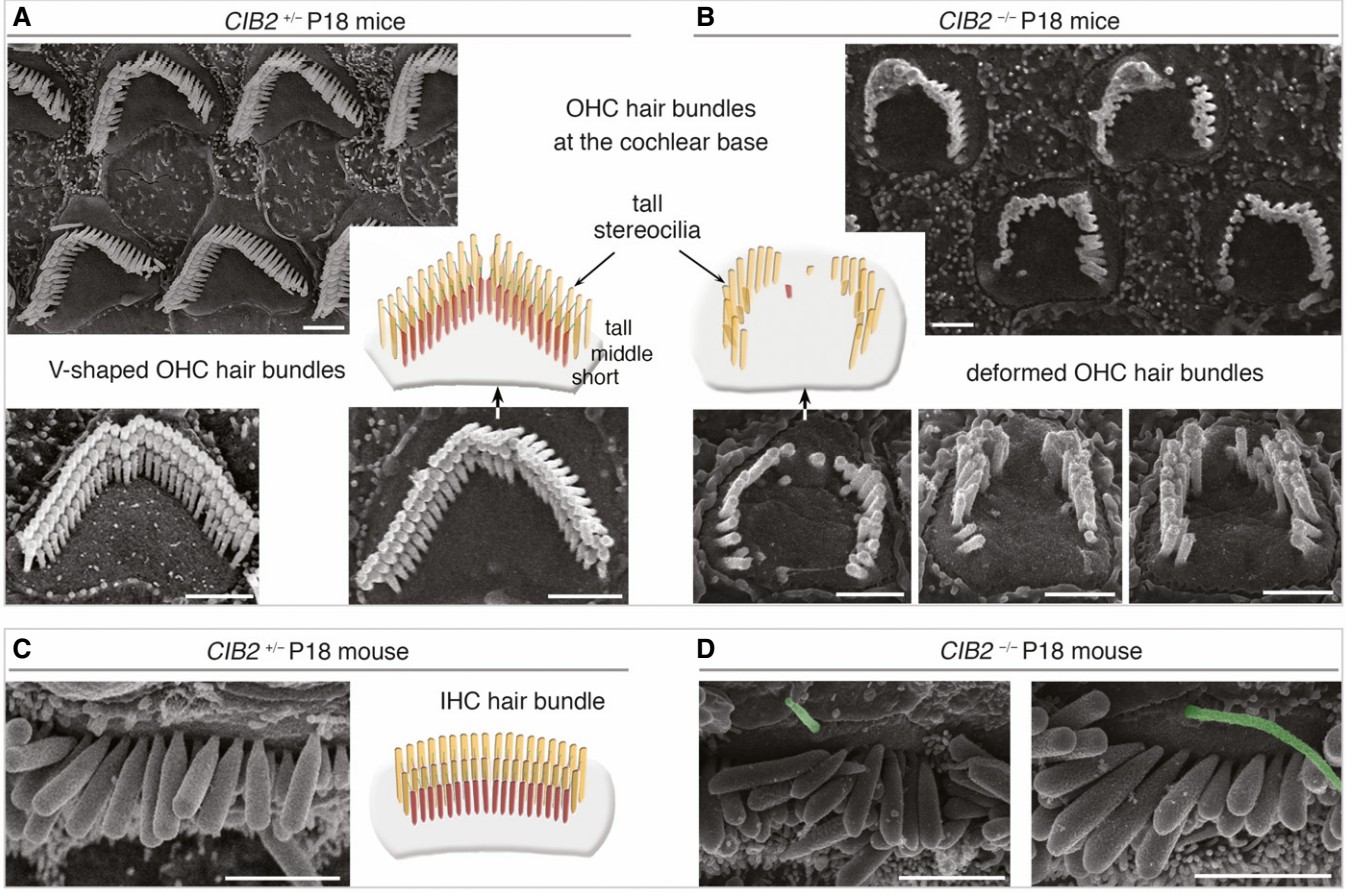

**Figure 4.  Deformation and rapid loss of stereocilia in *CIB2*$^{-/-}$ defective hair bundles.**

Top views of cochlear hair cells from *CIB2*$^{+/-}$ and *CIB2*$^{-/-}$ P18 mice.

A, B    Unlike in *CIB2*$^{+/-}$ mice (A), various abnormalities of the hair bundles are observed in the absence of CIB2: stereociliary shortening, total loss of short row stereocilia or split bundles.

C, D    The kinocilium (artificially coloured in green here), absent in *CIB2*$^{+/-}$ mice at this stage (C), persists abnormally in some IHC hair bundles of P18 *CIB2*$^{-/-}$ mice (D).

Data information: Scale bars: 2 μm.

---

severe balance defects consistently observed in mutant mice defective for any of the other five USH1 proteins.

## CIB2-interacting partners in the hair bundle stereocilia and at the apical surface of hair cells

CIB2 has been shown to interact with other USH proteins, including the USH1B myosin VIIa motor protein, and the USH2D scaffolding protein, whirlin (Riazuddin *et al*, 2012). Given the interdependence of USH proteins for the correct targeting of these proteins in the stereocilia (Boeda *et al*, 2002; Caberlotto *et al*, 2011), we explored the distribution of USH1/2 proteins in the hair bundles of *CIB2*$^{-/-}$ mice. We observed the typical enrichment in USH1 proteins—myosin VIIa, harmonin and protocadherin-15—in the apical region of the stereocilia in *CIB2*$^{-/-}$ mice, similar to that in control hair cells (Fig EV4A–C). We also assessed the expression of whirlin (USH2D), a component of the ankle links that has been shown to connect the basolateral regions of stereocilia during postnatal maturation of the hair bundle (Mburu *et al*, 2003; Michalski *et al*, 2007). Using an

anti-whirlin pan antibody recognizing the long and short whirlin splice isoforms (Ebrahim *et al*, 2016), we found that immunostaining localized to the basolateral regions and tips of the stereocilia in hair bundles from P7 control mice (Fig 7A). By contrast, in both the IHCs and OHCs of *CIB2*$^{-/-}$ mice, whirlin immunostaining was much weaker at the base of the stereocilia, but persisted at levels similar to those of control hair cells at the tips of stereocilia (Fig 7B).

We also investigated the impact of the loss of CIB2 on the distribution of integrins, the main CIB-interacting proteins (Figs 7C–F and EV4D). Integrins are heterodimeric transmembrane cell adhesion receptors consisting of non-covalently associated α- and β-subunits that form an essential link between the extracellular matrix and the intracellular cytoskeleton (Hynes, 2002; Hager *et al*, 2008). We investigated the distribution of integrin β1; we observed strong immunostaining restricted to the base of the stereocilia of hair bundles of OHCs and IHCS in P7 control mice. In IHCs, integrin β1 an immunostaining extending downwards in the taper region of the stereocilia was also observed (Fig EV4D, upper panels). Integrin β1 immunostaining was detectable in the IHC and OHC hair bundles of

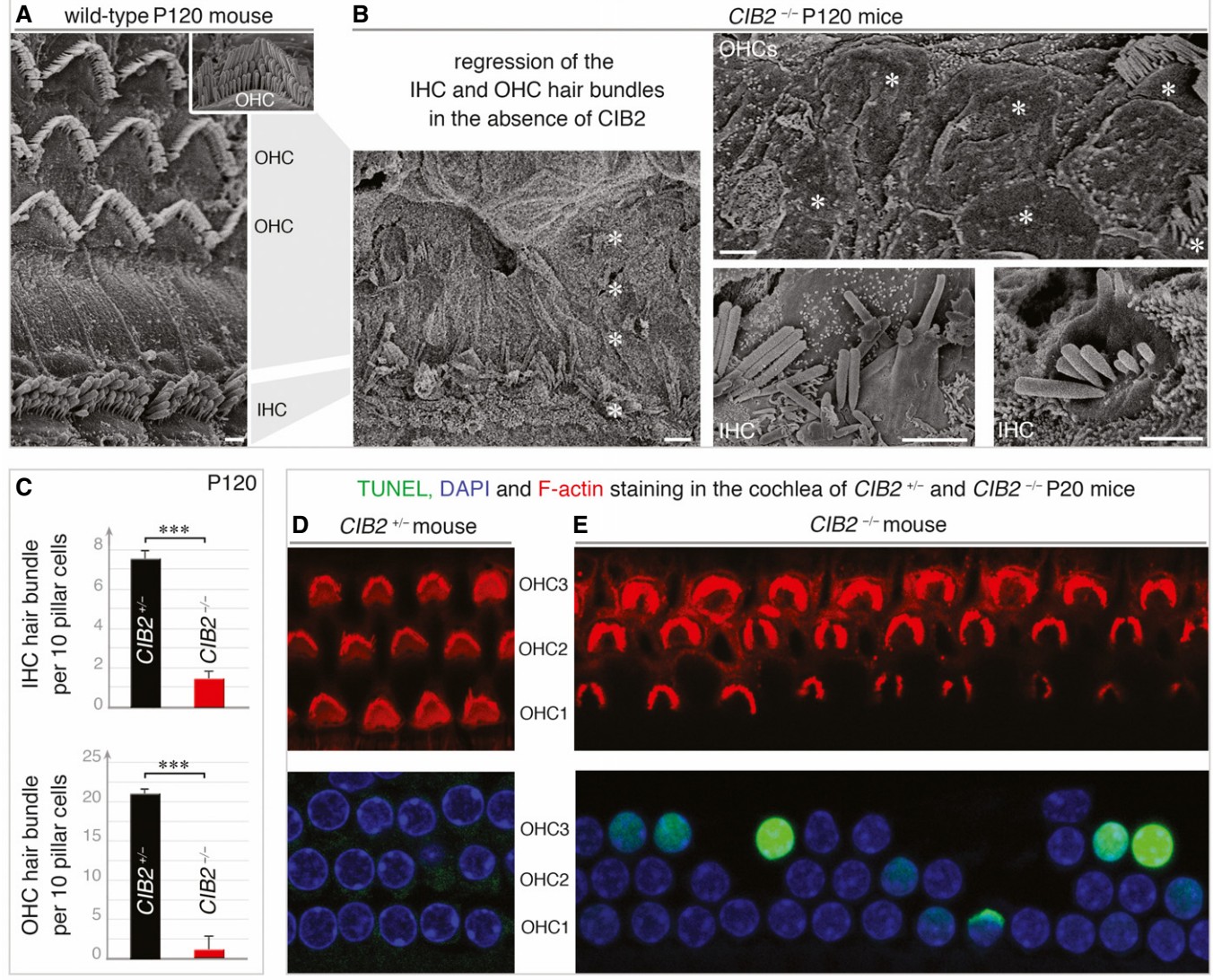

**Figure 5. Loss of cochlear hair bundles and hair cells in *CIB2*<sup>−/−</sup> mice.**

A, B Scanning electron microscopy micrographs of cochlear hair cells on P120. (A) The normal architecture of the sensory epithelium from *CIB2*$^{+/-}$ P120 mice is shown for comparison. (B) At this stage, most of the hair bundles have entirely disappeared at the apical surface of hair cells (asterisks). The persisting bundles are composed of few stereocilia, often fused or forming bleb-like structures, as shown in the mid-basal region of the cochlea in *CIB2*$^{-/-}$ P120 mice.

C Quantification of the number of IHC and OHC hair bundles present in this cochlear region in *CIB2*$^{+/-}$ (black) and *CIB2*$^{-/-}$ (red) mice. Nearly all the IHC (upper) and OHC (lower) hair bundles are lost on P120 mice in the absence of CIB2. Data (mean ± SEM) were analysed using an unpaired *t*-test with Welch's correction (***$P$ = 0.002 for IHC and OHC counts) ($n$ = 12 (region analysed) from 3 *CIB2*$^{+/-}$ mice, and $n$ = 12 from 4 *CIB2*$^{-/-}$ mice).

D, E Whole mounts of the cochlear sensory epithelia (mid-apical region) labelled with F-actin (red), DAPI (blue) and subjected to TUNEL staining (green) on P20 mice. Apoptotic hair cells are observed in the cochlea of *CIB2*$^{-/-}$ (E) but not *CIB2*$^{+/-}$ (D), mice.

Data information: Scale bars: 2 μm.

*CIB2*$^{-/-}$ P7 mice, though in IHCs the immunostaining seemed less restricted, extending further along the length of the stereocilia (Fig EV4D). The strong integrin α8 immunostaining detected mainly in the IHC and OHC cochlear hair bundles of P7 control mice was absent from the hair bundles of age-matched *CIB2*$^{-/-}$ mice (Fig 7C and D). These changes in localization were specific to the cochlea. In the vestibular organs, the integrin β1-immunostaining in the hair bundles, and that for integrin α8 in the supporting cells, was similar in *CIB2*$^{+/-}$ and *CIB2*$^{-/-}$ P7 mice (Figs 7E and F, and EV4D, lower panels).

Of the four CIB molecules, analyses of inner ear expression datasets revealed high levels of CIB1 and CIB2 in the cochlear hair cells (see http://shield.hms.harvard.edu; and http://gear.igs.umaryland.edu/). We therefore investigated the possible impact of CIB2 on the distribution of the CIB1 protein. On P7, CIB1 immunostaining was detected mainly at the apical surface of hair cells. In OHCs, it was observed in the kinocilium and in the basal body region at the periphery of the hair cell (Fig EV4E). However, this pattern of CIB1 immunostaining was modified in the OHCs of *CIB2*$^{-/-}$ mice, being relocated to the centre of the apical hair cell, over the cuticular plate (Fig EV4E).

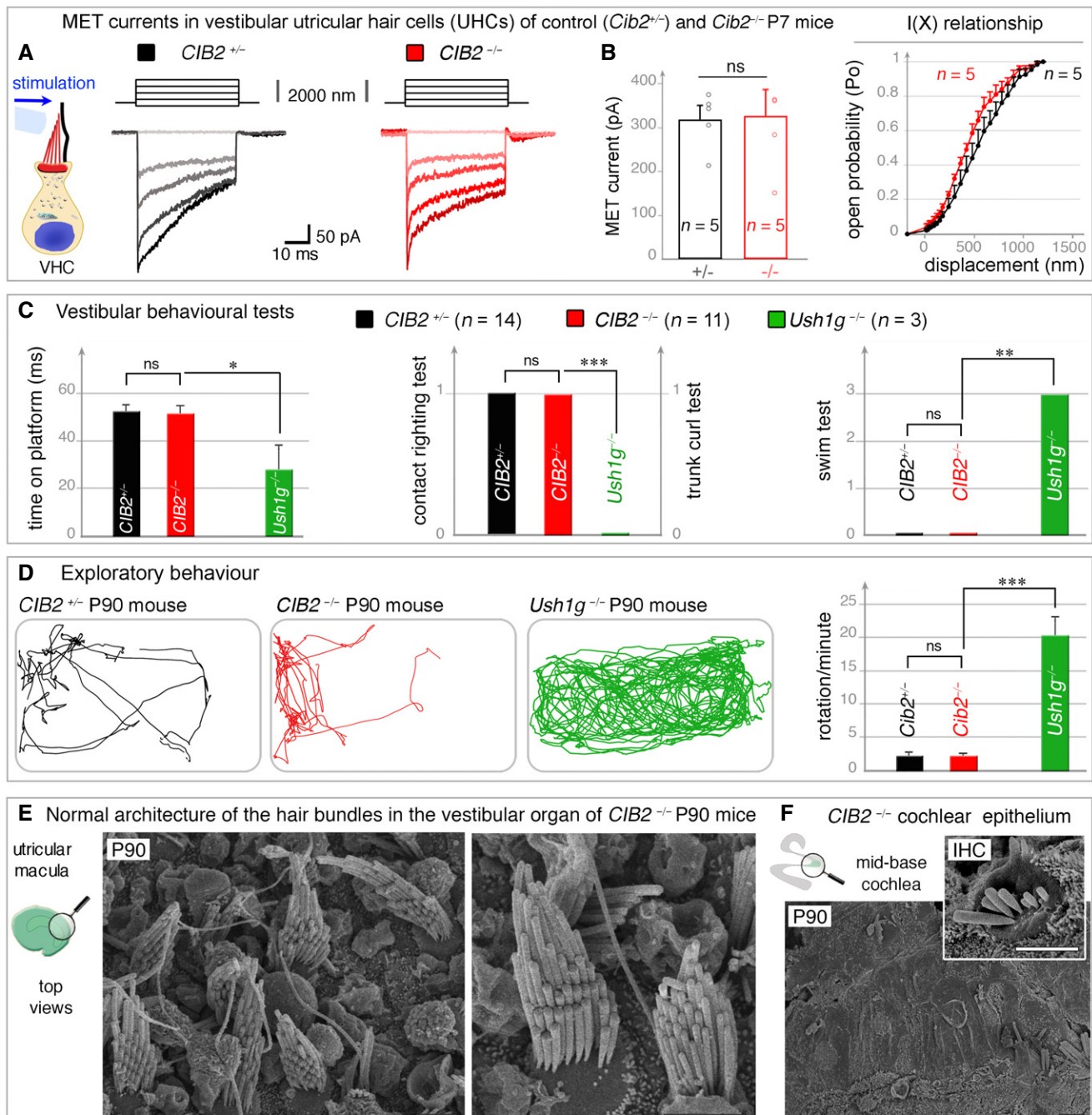

**Figure 6.  Absence of a vestibular phenotype in *CIB2*⁻/⁻ mice.**

A, B   The vestibular utricular hair cells (UHCs) of P7 *CIB2*⁻/⁻ mice have normal MET responses. (A) The panels on the left show the mechanical stimulation protocol, with examples of MET currents in control (black) and *CIB2*⁻/⁻ (dark-light red traces) UHCs. (B) The MET current values and the mean amplitude–displacement relationships (I(X), right panel) show near normal MET responses in *CIB2*⁻/⁻ (*n* = 5) UHCs, as compared to age-matched *CIB2*⁺/⁻ mice (*n* = 5) (mean ± SD). (Welch's *t*-test; "ns" indicates a statistically non-significant difference, *P* = 0.39).

C   Vestibular behavioural tests (platform, trunk–curl, contact–righting and swim tests). Unlike *Ush1g*⁻/⁻ mice (green, month 3, *n* = 3), the *CIB2*⁻/⁻ mice (red, month 3, *n* = 7, and month 7, *n* = 7) have no vestibular dysfunction, displaying similarly performances to age-matched control *CIB2*⁺/⁻ mice (black, month 3, *n* = 6 and month 7, *n* = 5) (mean ± SD). Being roughly similar, the values at months 3 and 7 were combined (Welch's *t*-test; *P* = 0.098 for platform test, ***P* = 0.007 for contact righting, and trunk curl tests, and **P* = 0.019 for swim test).

D   Representative open-field exploratory behaviour (2 min) by a 3-month-old mouse is shown for each genotype. Quantification of the number of rotations in 120 seconds (mean ± SEM), showing that, unlike *Ush1g*⁻/⁻ mice, which display the circling behaviour typical of USH1, *CIB2*⁻/⁻ mice have no vestibular defects (Welch's *t*-test; ***P* = 0.009 for rotations count) (*n* = 5  for *CIB2*⁺/⁻ and *CIB2*⁻/⁻ mice; and *n* = 3 for *Ush1g*⁻/⁻ mice).

E, F   Top views of UHCs from *CIB2*⁻/⁻ P90 mice, showing preserved and normally shaped hair bundles (E), contrasting with the severe loss at this stage of the hair bundles in the cochlea (F). Scale bars: 2 μm.

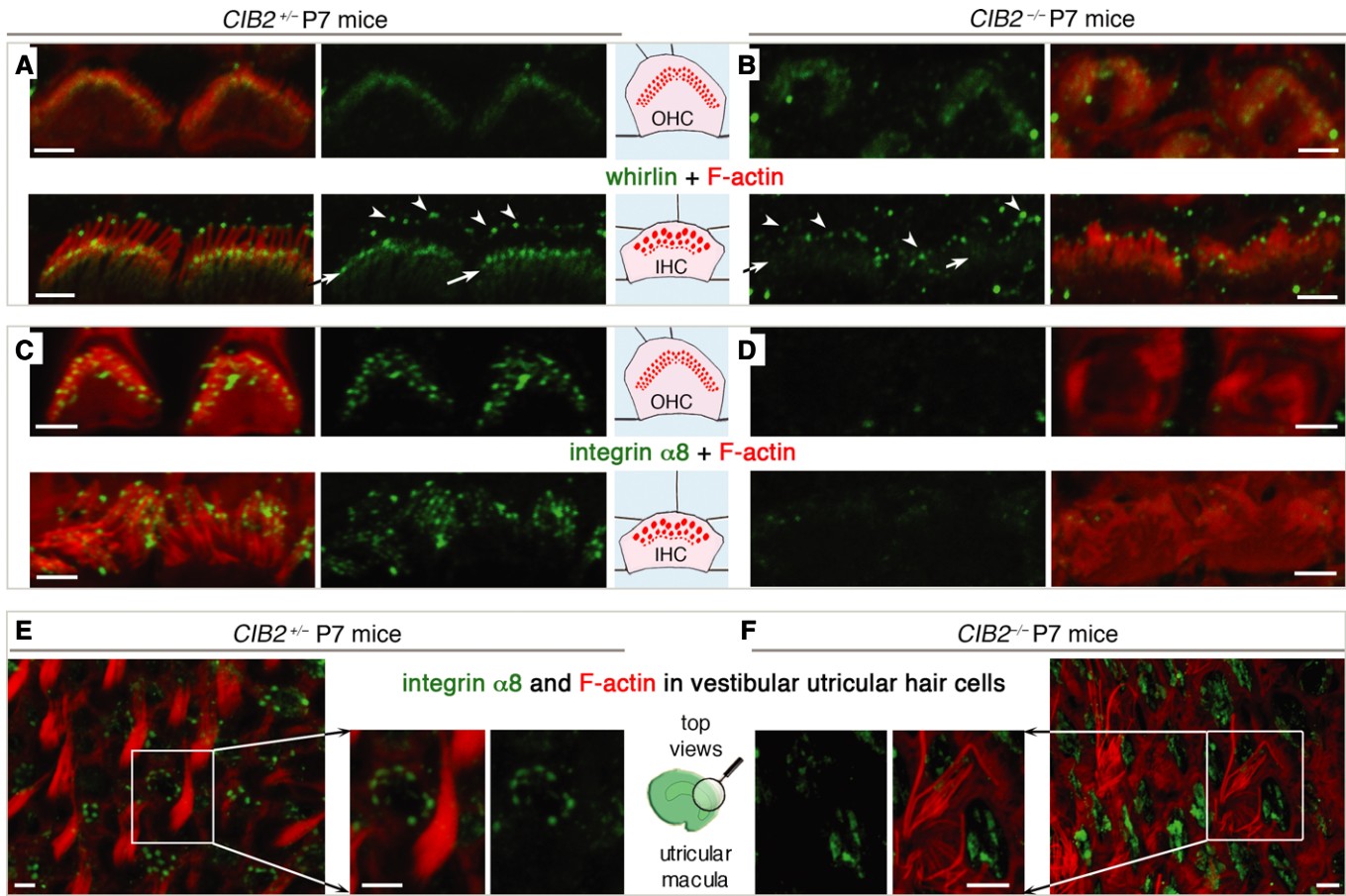

**Figure 7.   The distribution of whirlin and integrin α8 in CIB2$^{+/-}$ and CIB2$^{-/-}$ mice.**

A, B   In CIB2$^{+/-}$ P7 mice, whirlin immunostaining (green) is present in both the basolateral region (arrows) and the tips (arrowheads) of the stereocilia (A). In the absence of CIB2, whirlin immunostaining is weaker in the basolateral region, but not at the tips of the stereocilia (B).

C–F   Integrin α3 immunostaining in cochlear (C, D) and vestibular (E, F) hair cells. Integrin α8, which is detected along the length of the stereocilia of cochlear hair bundles in CIB2$^{+/-}$ mice (C), was absent from the cochlear hair bundles of mutant mice (D). In the vestibular epithelium on P7, integrin α8 immunostaining is localized at the apical surface of supporting cells, in both CIB2$^{+/-}$ (E) and CIB2$^{-/-}$ (F) mice.

Data information: Scale bars: 2 μm.

Together, our findings indicate that CIB2 plays a crucial role in the correct localization of membrane (integrin) and submembrane adaptor proteins (e.g. whirlin, CIB1) in the cochlear hair cells, notably at the base of the stereocilia.

### CIB2$^{-/-}$ mice have normal retinal function

We investigated other clinical features of Usher syndrome, by extending our analyses beyond the inner ear and assessing the retinal function of CIB2$^{-/-}$ mice on electroretinograms (ERGs; Fig 8A and B). We measured the retinal evoked potential responses characterized by an initial negative deflection (the a-wave) followed by a positive peak (the b-wave), the amplitudes of which vary with light intensity. Functional ERG measurements at month 3, under scotopic or photopic conditions, indicated that rod and cone functions were normal in these mutant mice (Fig 8A and B). The ERG responses were almost normal in shape, with unaffected time-to-peak values for the a- and b-waves (Fig 8A). The amplitudes measured at the peak of both the a- and b-waves were similar for control and

CIB2$^{-/-}$ mice (a-wave: 250.6 ± 15.2 and 253.9 ± 17.4, respectively; b-wave: 550.9 ± 26.3 and 443.9 ± 54.9, photopic ERG: 98.19 ± 9.6 and 73.95 ± 11.84; Fig 8B).

The overall laminar organization of the retina, examined on cryosections from mice aged three, nine and eleven months, was normal in CIB2$^{-/-}$ mice, with the retinal pigment epithelium cells and neuroretinal layers clearly distinguishable (see Fig 8C and D). By contrast to our findings for the cochlear epithelium (Fig 5E), no pycnotic nuclei indicative of degenerating cells were observed in any of the retinal cell layers, and TUNEL assays detected no apoptosis (Fig 8D). The normal distribution of Iba1-immunoreactive microglial cells, displaying typical resting state features (Fig 8E and F), also ruled out the possibility of reactive gliosis in young and old retinas from CIB2$^{-/-}$ mice. Focusing on the photoreceptor cell layer, we found no difference in the distribution of rhodopsin or red/green cone opsin, which were confined to the outer segments of rods and cones, respectively, in control CIB2$^{+/-}$ (Fig 8G) and CIB2$^{-/-}$ mice (Fig 8H). This finding was consistent with our histological and ultrastructural analyses for

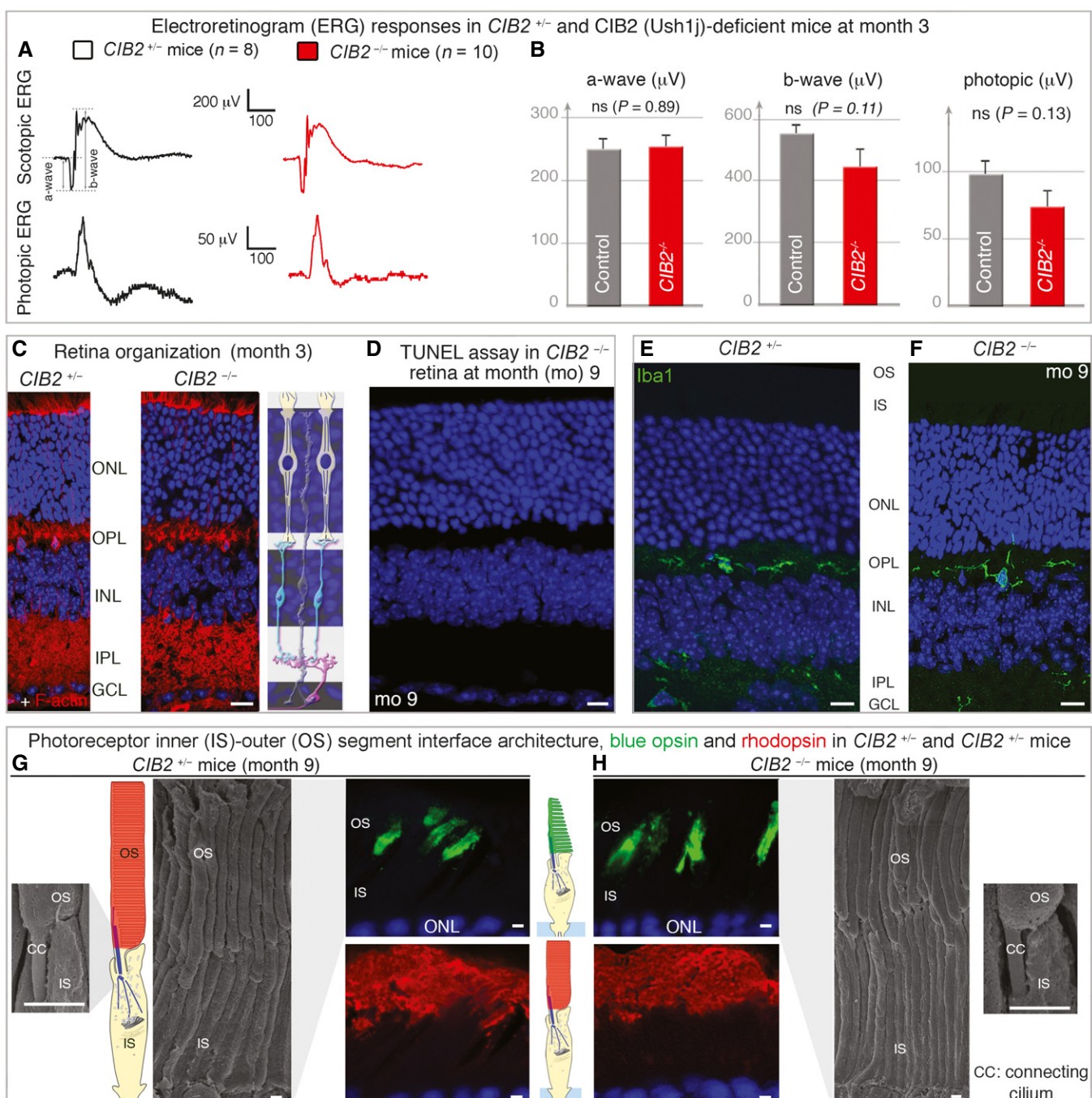

**Figure 8.  Absence of a retinal phenotype in *CIB2*$^{-/-}$ mice.**

A    Electroretinogram (ERG) recordings in 3-month-old *CIB2*$^{+/-}$ (black) and *CIB2*$^{-/-}$ (red) mice.

B    Quantification of scotopic a-wave (rod hyperpolarization) and b-wave (bipolar neuron depolarization) ERG amplitudes, and photopic ERG amplitudes (cone pathway), showing no significant difference between the two genotypes. The data shown are means ± SEM, *n* = 16 for *CIB2*$^{+/-}$ mice and *n* = 20 for *CIB2*$^{-/-}$ mice; Student's *t*-test; "ns" indicates a statistically non-significant difference, *P* > 0.1).

C, D   No difference in retinal layer organization was observed between *CIB2*$^{+/-}$ and *CIB2*$^{-/-}$ mice at 3 months as illustrated by DAPI and F-actin staining (C), and no TUNEL-positive cell was observed in *CIB2*$^{-/-}$ (red) mice at 9 months (D).

E, F   The Iba1-immunoreactive microglial cells are evenly distributed in the inner (IPL) and outer (OPL) plexiform layers in both *CIB2*$^{+/-}$ (E) and *CIB2*$^{-/-}$ (F) mice.

G, H   The scanning electron microscopy micrographs of the inner (IS)-outer (OS) segment interface show normal architecture in *CIB2*$^{+/-}$ (G) and *CIB2*$^{-/-}$ (H) mice at 9 months. The immunostaining for rhodopsin and blue opsin reveals no difference between the two genotypes, with the opsin staining normally restricted to unaffected outer segments.

Data information: Scale bars: 20 μm (C–F), 2 μm (G, H). ONL: outer nuclear layer, INL: inner nuclear layer, GCL: ganglion cell layer.

mice at the ages of 9 months, showing a normal architecture and organization of photoreceptor cells in $CIB2^{-/-}$ mice (Fig 8G and H).

Together, our molecular and morpho-functional findings provide no evidence for retinal dysfunction in the absence of CIB2. The lack of change in latency and amplitude responses in $CIB2^{-/-}$ mice indicate normal photoreceptor kinetics and no change in the sensitivity of photoreceptor cells.

### Identification of two new nonsense *CIB2* mutations in deaf patients with no indication for balance or retinal abnormalities

We then sought possible loss-of-function *CIB2* mutations in families in which detailed hearing, retinal and motor characterization had been performed. We identified two families from two ethnic backgrounds, Iranian (L-700) and Palestinian Arab (Trio-A), each with reported consanguinity (Fig 9A). Audiometric tests revealed bilateral symmetric prelingual severe-to-profound hearing loss across all frequencies in all affected individuals (Fig 9B, see also Appendix Tables S1 and S2). Fundoscopy ophthalmological evaluations revealed an absence of retinitis pigmentosa in both patients (patient II.1 aged 28 years and patient II.2 aged 26 years; Fig 9C). The affected individuals had normal motor development milestones, with no delays for sitting or walking, and further detailed physical and clinical examinations excluded syndromic features and suggesting the absence of balance defect (see also Appendix Tables S1). Targeted genomic enrichment and massively parallel sequencing with the OtoSCOPE® platform on probands from the two families yielded a mean of 10 million reads per sample and a coverage of 99.5 and 98.5% at ≥10× and ≥30×, respectively. After filtering for quality and MAF, a mean of nine variants per sample were identified. No copy number variation was detected in any of the samples. We filtered the variants under a recessive model, retaining only those that were homozygous or compound heterozygous. In the Trio-A family, a homozygous nonsense variant of *CIB2,* c.330T>A, was identified; this variant, located in exon 4, was predicted to produce a protein truncated at amino acid 110 (p.Tyr110*, located near the start of the second EF-hand domain) and to affect the coding sequences of all isoforms (Fig 9A). In family L-700, another homozygous nonsense variant, c.34C>T, was detected. This variant resulted in a premature stop codon at position 12 of the protein (p.Gln12*) and affected the coding sequence of isoforms CIB2-006, b and c (Fig 9A). Sanger sequencing confirmed the segregation of this variant with the deafness phenotype in the family.

Together, our findings show that, as in mice, null alleles of *CIB2* lead to profound hearing loss with no detectable balance or retinal dysfunction in humans.

## Discussion

Our findings show that MET responses, structural integrity of the hair bundles and hair-cell survival in the mature cochlea are critically dependent on CIB2, the protein defective in DFNB48 isolated deafness and in USH1J. Unlike the other five known USH1 proteins, however, functional CIB2 is not required to ensure the early cohesion and shaping of the growing/developing auditory hair bundle. It

is nonetheless required at the terminal differentiation and maturation stages. These features, together with the singular lack of vestibular deficits in $CIB2^{-/-}$ mice, raise questions about the possible role of CIB2 in Usher syndrome type I.

### CIB2, a key multifunctional protein in the cochlear auditory hair bundles

Our in-depth morpho-functional explorations of $CIB2^{-/-}$ mice suggest that the first few steps in hair-bundle morphogenesis are independent of *CIB2*. With normal hair bundle architecture in P6 $CIB2^{-/-}$ mice, the lack of CIB2 does not seem to impede the four stages of morphological development typically seen during the normal maturation of cochlear hair bundles in wild-type mice (Fig 4, see Fig EV5A): the initial production of stereocilia (stage I), migration of the kinocilium to the cell periphery and differential elongation of the rows of stereocilia (stage II), formation of the staircase pattern (stage III) and resorption of excess stereocilia (stage IV; Tilney *et al*, 1992; Kaltenbach *et al*, 1994; Goodyear *et al*, 2005; Nayak *et al*, 2007; Barr-Gillespie, 2015). However, despite the near normal appearance of the P6-P7 OHC hair bundles, no MET current responses were recorded in the OHCs of $CIB2^{-/-}$ mice on P7. The IHCs, as shown recently by Giese & colleagues, also display no MET current responses at P6 (Giese *et al*, 2017), showing the total lack of MET in the two types of auditory hair cells, which probably is related to its function within the TMC1/2 MET channel complex at the stereocilia tips of the two cell types (see also Fig EV5B). All in all, this lack of bundle function, the misshapen cochlear hair bundles and the ensuing rapid regression of stereocilia in the short and middle rows in the OHCs of $CIB2^{-/-}$ mice after P6-P7 together suggest a crucial role for CIB2 in the late steps of hair-bundle shaping and stereocilia architecture maintenance. During terminal maturation of the hair bundle, the kinocilium normally disappears by P12 in all auditory hair cells (Kikuchi & Hilding, 1965), but kinocilia were still observed, at least in IHCs, at mature stages, in P20 $CIB2^{-/-}$ mice. A similar persistence of the kinocilia has been reported for IHCs lacking myosin XVa, whirlin (Mustapha *et al*, 2007), or RFX transcription factors (Elkon *et al*, 2015), suggesting existence of potential pathways involving these proteins to allow normal kinocilia regression.

How does the lack of CIB2 lead to hair-bundle abnormalities? CIB2 was located in the hair bundle stereocilia and at the vicinity of the cuticular plate in the apical surface of the hair cells. Changes in the stereociliary distributions of whirlin and integrin, two CIB2-interacting partners in the auditory but not vestibular hair cells, as shown here for P7 hair cells, indicate a likely role for CIB2 as a molecular anchor for membrane and cytoskeleton-associated proteins in the stereocilia. Similar to endogenous CIB and integrin α2bβ3 that has been shown to translocate to the Triton X-100-insoluble cytoskeleton in aggregated platelets (Shock *et al*, 1999), CIB2 also, which probably exist in a multimeric form (A. Bahloul, unpublished), could associate with detergent-resistant microdomains in the stereocilia supporting its membrane to cytoskeleton cross-linking ability (see model Fig EV5C). The absence of CIB2 also leads to the relocation of CIB1 from the cell periphery to the apical cell centre, consistent with these two proteins having overlapping interdependent functions at the apical surface of hair cells. These changes are consistent with the findings of a recent study on T-cell lymphocytes,

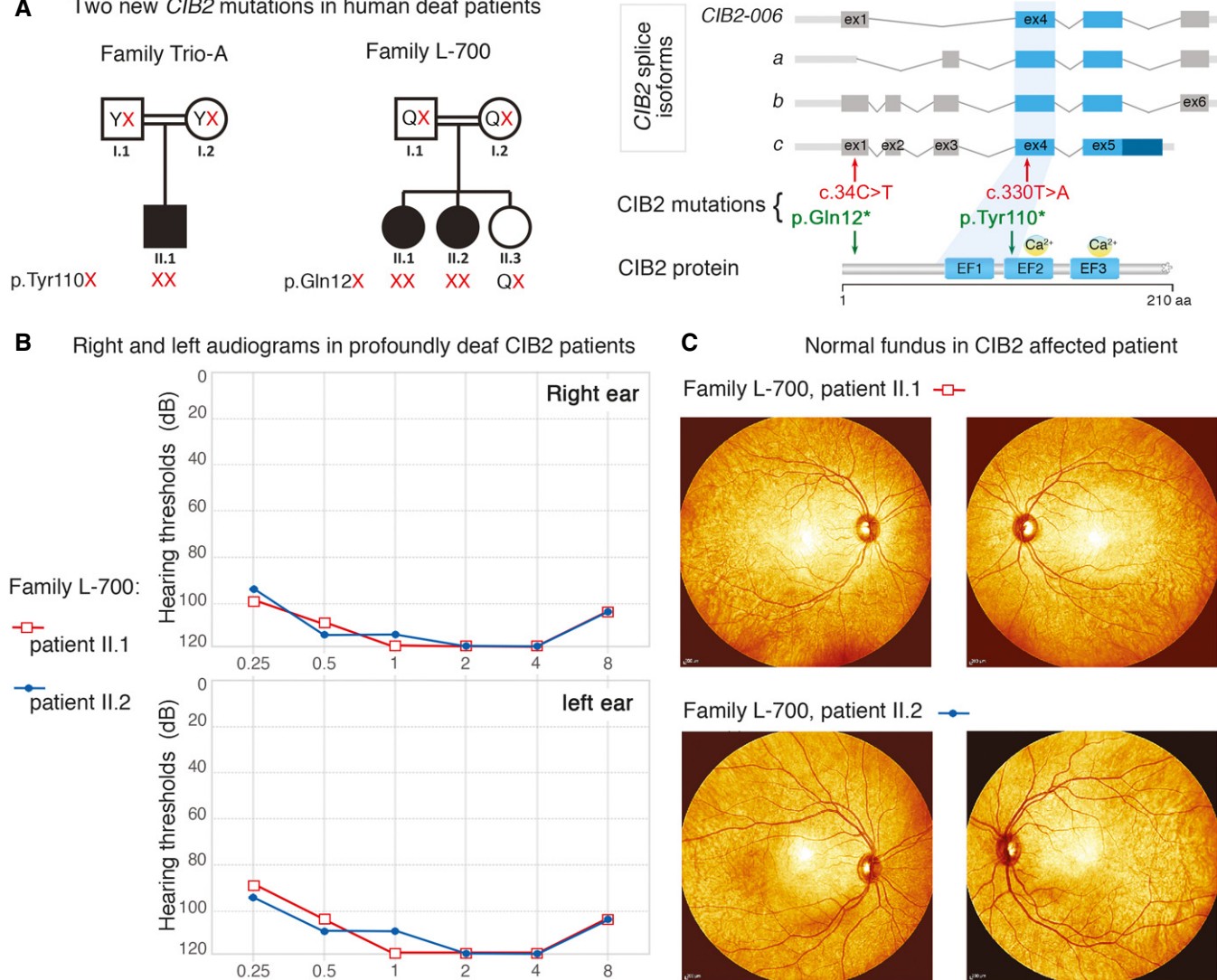

**Figure 9. Segregation of *CIB2* mutations, audiometric data and fundoscopy images in CIB2 patients.**

A   Pedigrees for the Trio-A (Palestinian Arab) and L-700 (Iranian) families. Filled symbols denote affected individuals, and double lines indicate consanguinity. Red letters represent the CIB2 mutant alleles segregating with the nonsyndromic hearing loss.

B   Audiograms were obtained using pure-tone audiometry with air conduction from frequencies from 250 Hz to 8,000 Hz. Severe-to-profound hearing loss was observed in CIB2 patients.

C   The fundoscopy images from CIB2 patients II.1 and II.2 (Family L-700) illustrate the normal architecture of the eye, with no pigment deposits indicative of a potential retinitis pigmentosa.

showing that both CIB1 and CIB2 proteins promoted the formation and stabilization of virological synapses (Godinho-Santos *et al*, 2016). The knockdown of CIB1 and/or CIB2 has been shown to impair the receptor-mediated entry of HIV into cells, associated with low levels of surface expression for integrin α4β7 and the membrane receptors CXCR4 and CCR5 (Godinho-Santos *et al*, 2016). Previous studies on laminin α2 chain-deficient skeletal muscle also reported a correlation between low levels of α7B integrin at the sarcolemma and low levels of CIB2 (Hager *et al*, 2008). Conversely, in transgenic mice overexpressing the laminin α1 chain, CIB2 expression is also partly reconstituted, consistent with interactions between CIB2 and integrin α7Bβ1D (Gawlik *et al*, 2006; Hager *et al*, 2008).

Together, our findings indicate that CIB2 probably functions as a component of a signalling platform influencing the integrin-mediated essential link between the extracellular matrix and the intracellular cytoskeleton of the hair bundle (see Fig EV5C). It has been suggested that CIB proteins activate integrin, in an "inside-out" signalling process promoting integrin ligand binding (Tsuboi, 2002; Freeman *et al*, 2013). The absence of CIB2 might therefore lead to weaker integrin activation on the stereocilia membrane, in turn affecting the "outside-in" signalling coupling the extracellular matrix (e.g. the α8 subunit) and intracellular responses in the hair bundle (e.g. whirlin and/or FERM proteins), thereby leading to a rapid regression of the hair bundles and hair-cell degeneration in *CIB2*−/− mice.

## CIB2, a key protein for the survival of cochlear auditory hair cells

One of our key findings is the important role played by CIB2 in auditory hair-cell survival. No apoptotic IHCs and OHCs were observed in P20 control mice, but large numbers of such cells were observed in age-matched *CIB2*$^{-/-}$ mice. It remains unclear how the absence of CIB2 leads to hair-cell apoptosis, but evidence suggesting a role for CIB proteins in these processes has been reported. In particular, CIB1 has been shown to function as a Ca$^{2+}$-sensitive negative regulator of ASK1 (apoptosis signal-regulating kinase-1)-mediated signalling events, involved in cell survival (Yoon *et al*, 2009), and CIB1 depletion in breast cancer cell lines results in significant cell death (Black *et al*, 2015). A model has been proposed in which CIB1 binds to ASK1, interfering with the binding of ASK1 to additional binding partners (e.g. TRAF2), thereby preventing the ASK1-mediated induction of cell death in response to reactive oxygen species, endoplasmic reticulum stress and other stimuli (Hattori *et al*, 2009; Kawarazaki *et al*, 2014). CIB2 may play a similar role in the maintenance of auditory hair-cell viability. An analysis of an RNA transcript profiling dataset revealed that ASK1 (also called map3 kinase 5, MAP3K5) was also produced in the sensory hair cells (http://gear.igs.umaryland.edu/). A role in age-related hearing loss has been proposed for ASK1 in the auditory cortex, with a decrease in the interaction between thioredoxin-2 and ASK1 thought to be responsible for the observed neuronal apoptosis (Dinh *et al*, 2015; Sun *et al*, 2015). Also, a recent study showed that similar to CIB1, CIB2 negatively regulates oncogenic signalling in ovarian cancer via its interaction with sphingosine kinase 1 (SK1), a regulator of the cellular balance between pro-apoptotic and pro-survival sphingolipids (Zhu *et al*, 2017). It remains possible that the absence of CIB2 triggers stress-induced ASK1 activation and/or interferes with translocation of SK1 to cell plasma membrane, leading to an upregulation of pro-apoptotic proteins, leading to the death of auditory hair cells.

## CIB2: isolated deafness and the Usher syndrome phenotype

The profound deafness observed in *CIB2*$^{-/-}$ mice, lacking a functional CIB2 protein for all four *Cib2* splice isoforms (Fig 1A, see also Giese *et al*, 2017; Zou *et al*, 2017), is reminiscent of the prelingual severe-to-profound hearing loss observed in DFNB48-affected patients (Riazuddin *et al*, 2012; Patel *et al*, 2015; Seco *et al*, 2016; and this study). However, none of the cochlear phenotypes associated with the lack of CIB2 described above were observed in CIB2-defective vestibular organs. The normal functioning of CIB2-deficient balance organs was established by comparative behaviour tests (e.g. rotarod, platform and swimming tests, or open-field exploratory behaviour). TUNEL assays performed at the ages of one and 6 months revealed an absence of apoptotic hair cells, and vestibular hair cells had a typical organization, regardless of the stage analysed. This organization persisted even at late adult stages when all the cochlear hair bundles had disappeared and the cochlear hair cells had degenerated (Fig 6E and F). By contrast to the results obtained for the cochlea, no significant change in integrin immunostaining was observed in the vestibular hair bundle (Figs 7E and F, and EV4C). The opposite pattern has already been reported in α8β1-deficient mice (analysed on embryonic day 16 (E16) and at birth), which display altered immunostaining for fibronectin and collagen IV over hair cells in the vestibular utricular epithelium, but not in the

cochlea (Littlewood Evans & Muller, 2000). It has been suggested that a potential integrin code, differing between the hair cells of the different inner ear sensory compartments, the cochlea and the five vestibular sensory epithelia, regulates the hair-cell environment.

Differences in the functional redundancy of CIB paralogs between cochlear and vestibular hair cells may account for the phenotypic discrepancies described above. Under similar immunofluorescence conditions, we could detect CIB2 in the cochlear, but not vestibular, hair cells (Fig 2A and D), consistent with analyses of RNAseq expression datasets showing that *CIB2* transcript levels are four times higher in P4 or P16 cochlear hair cells than in vestibular hair cells (Scheffer *et al*, 2015, see expression databases, shield.hms.harvard.edu; and http://gear.igs.umaryland.edu/). Compensation between CIB family members in pathogenic conditions has been demonstrated in *CIB1*$^{-/-}$ mice (Denofrio *et al*, 2008). Indeed, it has been suggested that the higher levels of CIB3 mRNA in cultured megakaryocytes isolated from *CIB1*$^{-/-}$ mice (Yuan *et al*, 2006) compensate for the loss of *CIB1*, probably accounting for the lack of overt defects of platelet function in *CIB1*$^{-/-}$ mice (Denofrio *et al*, 2008). Using quantitative RT–PCR to investigate the expression of *CIB* transcripts (*CIB 1 to 4*) in the inner ear at P12, Giese and co-workers found levels of *CIB3* expression in the vestibular sensory epithelia to be almost eight folds those in the cochlea (Giese *et al*, 2017). In the absence of CIB2, a significant upregulation was observed only for CIB1, and this upregulation was specific to the cochlea (Giese *et al*, 2017). These data suggest that CIB1 plays a role at least partially different from that of CIB2 in the cochlea and indicate that CIB3 may be crucial for vestibular function. Further studies are required to unravel the precise role of CIB paralogs in the different inner ear compartments.

The lack of vestibular deficits, whatever the stage analysed, and the late occurrence of hair-bundle abnormalities, leading to a rapid regression of the stereocilia and the degeneration of most OHCs and IHCs by the age of 4 months in the absence of CIB2, clearly single out *CIB2*$^{-/-}$ mice as different from other Ush1 mutant mice. Genome analyses of the retina in *Drosophila* identified one CIB-related gene, *CG9236*, referred to as the *cib2* ortholog (Riazuddin *et al*, 2012). Knockdown of this gene in *Drosophila* by RNA interference (RNAi) led to a decrease in the amplitude of photoresponses, indicating a key role for CIB in the retina of this species (Riazuddin *et al*, 2012). By contrast, ERG measurements in *CIB2*$^{-/-}$ mice at the age of 3–4 months detected no retinal defect, and histological and molecular analyses at the ages of 7 and 11 months revealed normal layer organization, correct photoreceptor architecture, normal targeting of opsin molecules and an absence of degenerative or inflammation processes (see Fig 8). Interspecies differences in the functional redundancy of CIB proteins and/or differences in photo-transduction cascades may explain the presence of retinal defects in *Drosophila* (Riazuddin *et al*, 2012), and their absence in mice (this study). In the fly, opsin stimulation leads to the activation of a phospholipase Cβ (PLC), triggering the opening of two classes of Ca$^{2+}$-permeable light-sensitive channels: the transient receptor potential (TRP) and TRP-like (TRPL) cation channels. By contrast, in mice, the stimulation of opsin molecules in the rods and cones leads to an increase in the activity of a cyclic guanosine monophosphate (cGMP) phosphodiesterase, resulting in a decrease in cGMP levels, and closure of the cGMP-gated channels (Wang & Montell, 2007). Studies of *CIB2* expression in the retina revealed no unambiguous difference

in labelling between wild-type and $CIB2^{-/-}$ mice (see Appendix Fig S1). We found no $CIB2$ immunostaining in the calyceal processes of macaque photoreceptor cells (such as that observed for USH1 proteins, Sahly *et al*, 2012) or in the periciliary ridge of mouse and macaque photoreceptor cells (such as that observed for USH2 proteins, Sahly *et al*, 2012; see Appendix Fig S1). These expression studies therefore provided no support for a potential link between CIB2 and Usher syndrome. Finally, following on from the initial identification of $CIB2$ mutations in Pakistani and Turkish families (Riazuddin *et al*, 2012), new $CIB2$ mutations have been found in Dutch, Caribbean Hispanic (Patel *et al*, 2015; Seco *et al*, 2016) and Iranian and Palestinian Arab (this study, see Fig 9A–C) families. Of all the nine mutations of $CIB2$ described to date (including those described here), only the p.Glu64Asp missense variant, described in one Pakistani family, has been reported to cause USH1J (Riazuddin *et al*, 2012). Like the p.Glu64Asp USH1J mutation, two missense mutations, p.Arg66Trp located two amino acids downstream from the USH1J variant and p.Phe91Ser located in the first EF domain, have also been reported to affect the interaction of CIB2 with integrin, without interfering with CIB2 protein levels or targeting to the stereocilia (Riazuddin *et al*, 2012; Patel *et al*, 2015; Seco *et al*, 2016). Nevertheless, all patients with the p.Arg66Trp or p.Phe91Ser mutation have the DFNB48 form of isolated deafness, with no signs of retinitis pigmentosa or vestibular dysfunction (Riazuddin *et al*, 2012; Patel *et al*, 2015; Seco *et al*, 2016). Recent sequencing studies on 427 patients with USH led to the identification of biallelic or monoallelic mutations of Usher genes in 421 individuals with USH, but no $CIB2$ mutations were detected in any of these patients (Bonnet *et al*, 2016). Further studies are required to determine whether and how the p.Glu64Asp mutation causes phenotypic abnormalities affecting hearing, balance, and vision.

In summary, our findings indicate that CIB2 is necessary for proper function and maintenance of the auditory hair cells, with the protein being critically involved in the mechanoelectrical transduction activity of the hair bundle, the maintenance of its structural integrity and hair-cell survival. Consistently, loss-of-function mutations of $CIB2$ in humans also lead to profound deafness without vestibular or retinal dysfunction. Therefore, caution is required when providing genetic counselling (Idan *et al*, 2013; Koffler *et al*, 2015) for families with patients carrying $CIB2$ mutations, as these mutations may not necessarily lead to Usher syndrome.

# Materials and Methods

## Animals and ethics statements

The study protocol was approved by the local ethics committees and was carried out following the ethical principles for medical research involving human subjects defined by the WMA Declaration of Helsinki and the Department of Health and Human Services Belmont Report. Informed written consent was obtained from every participant. Animals were housed in the Institut Pasteur animal facilities accredited by the French Ministry of Agriculture to perform experiments on live mice (accreditation 75-15-01, issued on 6 September 2013 in appliance of the French and European regulations on care and protection of the Laboratory Animals (EC Directive 2010/63, French Law 2013-118, 6 February 2013). Mice were housed in cages with wood shaving as bedding, and enriched with wool cotton balls, and were granted unlimited access to water and food. Morphological and functional analyses were performed at different time points from birth to 11 months of age as mentioned in the text and figure legends; females and males were used indiscriminately in the study. The corresponding author confirms that protocols were approved by the veterinary staff of the Institut Pasteur animal facility and were performed in compliance with the NIH Animal Welfare Insurance #A5476-01 issued on 31 July 2012.

## Generation of knockout mice for CIB2

We generated a CIB2-deficient mouse strain in a C57BL/6N genetic background. First, we engineered a $CIB2^{fl/fl}$ mouse by floxing exon 4 of $CIB2$. These $CIB2^{fl/fl}$ mice were crossed with $PGK\text{-}cre^{+/-}$ mice to achieve the constitutive ubiquitous inactivation of CIB2 ($CIB2^{-/-}$ mice).

The genotype of $CIB2^{-/-}$ mice was determined by two PCR amplifications: one using oligo-Int3-15019 (5′-ggaacagaggcagatgctgggaa-3′) and oligo-Int3-22494 (5′-gcagccatcacttagggtaggaaagg-3′) to detect the wild allele and one using oligo-Int3-15019 and oligo-Int4-22987 (5′-ctctcacagacccctcccaccacatcc-3′) to detect a deleted allele.

For RT–PCR analyses, fresh tissues were collected and quickly frozen in liquid nitrogen and stored at −80°C until processing. Total RNA was isolated from the inner ear, eye, brain, muscle, kidney and testis of the two genotypes (wild-type and $CIB2^{-/-}$ mice) with TRIzol Reagent (Invitrogen) according to the manufacturer's instructions. Total RNA (400 ng) was reverse-transcribed with the Superscript One-Step RT–PCR system (Invitrogen) using various gene-specific primers: for amplicons between exon 1 and exon 4: forward exon 1: TGGGGAACAAGCAGACCATCTTCAC and reverse exon 4: ATAGATCTTGAAGGCATAGTTTGCC; for amplicons between the junction of exons 1/2 and that of exons 5/6: forward exon 1/2: TGGACAACTACCAGGACTGCACTTTC and reverse exon 5/6: TCGAATGTGGAAGGTGCTGAG (see Fig 1A and B). Reverse transcription negative controls, containing all RT–PCR reagents except the reverse transcriptase, were used; no amplification was observed. As an endogenous control (see Fig 1B), we used primers derived against mouse β-actin: forward: ACCTGACAGAC-TACCTCAT; and reverse: AGACAGCACTGTGTTGGCAT.

## *In vivo* auditory tests

We assessed the hearing sensitivity of the mice by recording auditory brainstem responses (ABRs), cochlear microphonic (CM) electrical potential and distortion product otoacoustic emissions (DPOAEs) in anesthetized mice and analysing them as previously described (Le Calvez *et al*, 1998; Kamiya *et al*, 2014). Mice were anesthetized by intraperitoneal injections of a mixture of levomepromazine (2 mg/kg) and ketamine (150 mg/kg), and their body temperature was maintained at 37°C with a thermostatically controlled heating pad. Auditory brainstem responses (ABRs) were recorded with stainless steel electrodes applied to the vertex and ipsilateral mastoid, with the lower back serving as the ground, as previously described (Le Calvez *et al*, 1998; Kamiya *et al*, 2014). The sound stimuli were pure tones at 5, 10, 15, 20, 32 and 40 kHz, each tested at SPL levels between 10 and 115 dB. ABR thresholds were identified as the weakest stimulus producing a detectable

electroencephalogram. DPOAEs were studied with a Cub$^e$Dis system (Mimosa Acoustics; ER10B microphone, Etymotic Research). Two pure-tone stimuli at frequencies f1 and f2 were emitted simultaneously and at the same SPL, from 20 to 70 dB in 10 dB steps and then to 75 dB SPL. The f2 frequency was swept from 5 to 32 kHz with increases of 1/8$^{th}$ of an octave, constantly varying f1 to maintain an f2/f1 ratio of 1.20. Only the cubic difference tone at 2f1–f2, the most prominent tone produced by mammalian ears, was measured (Le Calvez et al, 1998; Kamiya et al, 2014). DPOAE thresholds were defined as the lowest SPL resulting in a DPOAE significantly above background noise.

## MET current recordings

For electrophysiological recordings, the inner ears of P7 mice were finely dissected, and cochlear and utricular explants were bathed in an extracellular solution containing 146 mM NaCl, 5.8 mM KCl, 1.5 mM CaCl$_2$, 0.7 mM NaH$_2$PO$_4$, 2 mM sodium pyruvate, 10 mM glucose and 10 mM HEPES, pH 7.4 (305 mOsm/kg; Lelli et al, 2016). Borosilicate patch pipettes (1–2 MΩ) were filled with an intracellular recording solution containing 130 mM KCl, 10 mM NaCl, 3.5 mM MgCl$_2$, 1 mM EGTA, 5 mM potassium ATP, 0.5 mM GTP and 5 mM HEPES, pH 7.4 (290 mOsm/kg). The Ca$^{2+}$ currents of OHCs lying in the apical third of the cochlea and of UHCs were recorded in the whole-cell voltage-clamp configuration at −80 mV and 20–25°C, with an EPC-10 patch-clamp amplifier and Patchmaster software (Heka Elektronik, Lambrecht, Germany). No correction was made for liquid junction potential. Series resistance was systematically <10 MΩ and was compensated to 70%. A rigid fire-polished glass rod with a tip diameter of 2–3 μm was used to displace the hair bundles mechanically. The probe was secured to a piezoelectric actuator (PA8/12; Piezosystem Jena) driven by a fast voltage amplifier (ENV800; Piezosystem Jena). The resonance of the actuator was limited by subjecting the stimulating signal of the EPC-10 amplifier to low band-pass filtering at 5 kHz with a Bessel four-pole filter (model 3362; Krohn-Kite). Each MET current reported is the transducer response averaged over five consecutive stimulations (Lelli et al, 2016).

## Vestibular behavioural tests

We used a battery of behavioural tests to assess the vestibular function of CIB2$^{−/−}$ mice, as previously described (Hardisty-Hughes et al, 2010). In the platform test, the specimen was positioned on a small platform (7 × 7 cm) at a height of 29 cm, and the number of falls from the platform was counted over one minute. In the suspension test, we held the mouse by its tail and observed if it managed to reach a horizontal landing surface (score 1) and curl its trunk towards its tail (score 1) or not (score 0). The contact righting test consisted in placing a mouse in a closed transparent tube and determining whether it was able to regain standing position upon an 180° rotation of the tube (score 1). The swimming performance was assessed by placing each mouse in a container filled with water at 22–23°C (score 0 = mice swim, score 1 = irregular swim, score 2 = immobile floating, score 3 = underwater tumbling). Finally, we evaluated hyperactive and circling behaviours using a tracking software (Etho-vision de Noldus Information Technology, Wageningen, the Netherlands) to count the number of clockwise and anti-clockwise turns over a 120-s stay in an open-field chamber (37 × 18.5 cm).

## Electroretinography

Animals were allowed to adapt to darkness overnight. They were then anesthetized with a mixture of ketamine (80 mg/kg, Axience, France) and xylazine (8 mg/kg, Axience, France) and laid over a heating pad to maintain their body temperature at 37°C. Their pupils were dilated with tropicamide (Mydriaticum; Théa, Clermont-Ferrand, France) and phenylephrine (Neosynephrine; Europhta, Monaco). The cornea was locally anesthetized with oxybuprocaïne chlorhydrate (Théa, Clermont-Ferrand, France). Upper and lower lids were retracted to keep eyes open and bulging. Retinal responses were recorded with a gold-loop electrode brought into contact with the cornea through a layer of lubrithal (Dechra, France), with needle electrodes placed in the cheeks and back used as reference and ground electrodes, respectively (Yang et al, 2009). The light stimuli were provided by a Led in a Ganzfeld stimulator (SIEM Bio-médicale, France). Responses were amplified and filtered (1 Hz-low and 300 Hz-high cut-off filters) with a one-channel DC-/AC amplifier. One level of stimulus intensity (8 cd.s/m$^2$) was used for scotopic ERG recording. Each of the responses presented was averaged over five flash stimulations. Photopic cone ERGs were recorded in a rod-suppressing background light of 20 cd/m$^2$, after a five-minute adaptation period. A 8 cd.s/m$^2$ level of stimulus intensity was used for the light-adapted ERGs. Each cone photopic ERG response presented was averaged over ten consecutive flashes.

## Scanning electron microscopy

For scanning electron microscopy, dissected inner ears and isolated eyes were fixed by incubation in 2.5% glutaraldehyde in 0.1 M cacodylate buffer for 2 h at room temperature. The samples were washed in 0.1 M cacodylate then several times in water, processed by the osmium tetroxide/thiocarbohydrazide (OTOTO) impregnation method and progressively dehydrated by incubation in increasing concentrations of ethanol, as previously described (Furness et al, 2008). Dried samples were analysed with a field emission scanning electron microscope (Jeol JSM6700F operating at 5 kV). Images were obtained with a charge-coupled device camera (SIS Megaview3; Surface Imaging Systems), equipped with analySIS (Soft Imaging System) and were processed with Photoshop CS6. Hair bundle counts were performed using scanning electron microscopy micrographs in the mid-apical region of 4-month-old control (n = 3) and CIB2$^{−/−}$ mice (n = 4), counting the number of adjacent IHCs and OHCs to ten pillar cells.

## Immunofluorescence experiments

For immunofluorescence on whole-mount preparations, inner ear explants were dissected from the temporal bone and fixed by incubation in either 2% paraformaldehyde (PFA) in phosphate-buffered saline (PBS) pH 7.4 at 4°C overnight or in 4% PFA at room temperature for 30 min for later stages (Legendre et al, 2008). Samples were blocked by incubation in PBS supplemented with 20% normal goat serum (NGS) and 0.3% Triton X-100 at room temperature for 1 h. Samples were incubated overnight with primary antibodies in PBS 2% bovine serum albumin (BSA) at 4°C. Samples were incubated with specific secondary antibodies and with phalloidin for actin

staining, when required, at room temperature for 1 h. They were then immersed in DAPI (Sigma) for nuclear labelling. Samples were mounted in Fluorsave (Calbiochem, La Jolla, CA).

For immunofluorescence studies on cryosections, eyes from mouse or macaque were fixed in PBS 4% PFA for 30 min at 4°C. Macaque eyes were collected from adult cynomolgus monkeys (*Macaca fascicularis*) housed at the MIRcen platform (CEA/INSERM, Fontenay-aux-Roses, France). These animals were killed as controls in other unrelated experiments. After several washes in PBS, the samples were infused with a 25% wt/vol sucrose solution over 12 h at 4°C, then immersed in OCT medium (Tissue-Tek) and frozen in dry ice and stored at −80°C until use (Papal *et al*, 2013). Cryosections (10 μm) were rehydrated, permeabilized by incubation in PBS 0.1% Triton X-100 for 5 min, washed thoroughly in PBS and then blocked by incubation in PBS 1% BSA and 0.05% Tween-20 at room temperature for 1 h. Primary antibodies were diluted in blocking buffer and then incubated with the samples overnight at 4°C. Samples were then processed as described for whole mounts.

We checked for apoptosis in the inner ears and eyes of control and *CIB2*$^{-/-}$ mice, by incubating samples with phalloidin to label actin filaments, then washing them and subjecting them to processing with the *in situ* Cell Death Detection Kit, Fluorescein (Roche), according to the manufacturer's instructions.

For CIB2, we used a commercially available rabbit polyclonal anti-CIB2 antibody (ab111908, Abcam) and a newly generated rabbit polyclonal antibody directed against human CIB2. We used the peptide "ELTLARLTKSELDEEEVVLVCDKVI" (aa 129–153, NP_006374.1) as this sequence is not shared with CIB1 protein. Antibodies were affinity-purified from immune sera (obtained from AgroBio, France) using the peptide antigen coupled to an NHS column (GE Healthcare). We checked the specificity of the antibodies produced by immunofluorescence experiments in which CIB2$^{-/-}$ samples were used as negative controls.

To detect USH1 and USH2 proteins, we used the rabbit polyclonal antibodies directed against myosin VIIa (1:200), harmonin (1:100), protocadherin-15 (1:100), Adgvr1 (1:80) and whirlin (1:100; Sahly *et al*, 2012). Other primary antibodies used were rabbit polyclonal anti blue cone opsin (1:250; AB5407, Merck-Millipore), Iba1 (1:500; 019-19741, Wako Chemicals), integrin α8 (1:100; ABT139, Merck-Millipore), integrin β1 (1:100; sc_9970, Santa-Cruz), CIB1 (1:100; ab198845, Abcam) and the mouse monoclonal anti-rhodopsin (1:500; MAB5316, Merck-Millipore). The specific secondary antibodies were the Sigma-Aldrich ATTO 488-conjugated goat anti–rabbit IgG antibody (1:500) and ATTO 565 goat anti-mouse IgG antibody (1:500). ATTO 565 phalloidin (1:200; Sigma-Aldrich) was used to label F-actin. Samples were imaged at room temperature with a confocal microscope (LSM 700; Carl Zeiss) fitted with a Plan-Apochromat ×63 NA 1.4 oil immersion objective from Carl Zeiss.

## Subjects and clinical evaluations

One Iranian family (L-700) and one Palestinian family (Trio-A) segregating ARNSHL were obtained for this study. Clinical evaluation was completed in all affected persons by a clinical geneticist, otolaryngologist and ophthalmologist. Hearing thresholds were measured by pure-tone audiometry at 0.25, 0.5, 1, 2, 3, 4 and 8 kHz. After obtaining written informed consent to participate in

this study, blood samples were collected from all family members and genomic DNA (gDNA) was extracted. All procedures were approved by the human research Institutional Review Boards at the Welfare Science and Rehabilitation University and the Iran University of Medical Sciences, Tehran (Iran), the University of Iowa, Iowa City, Iowa (USA).

## Targeted genomic enrichment, massively parallel sequencing and bioinformatic analysis

Targeted genomic enrichment (TGE) and massively parallel sequencing (MPS) using the OtoSCOPE® platform was performed on the proband in Trio-A and one affected individual from family L-700, to screen all known genes implicated in NSHL and USH for possible mutations as previously described (Booth *et al*, 2015; Sloan-Heggen *et al*, 2016). Enriched libraries were sequenced at the University of Iowa Genomics Core on an Illumina HiSeq 2000 (Illumina, Inc., San Diego, CA) using 100-bp paired-end reads. Data analysis was performed using a custom Galaxy annotation pipeline as previously described (Azaiez, *et al*, 2014, 2015; Booth *et al*, 2015). After mapping and variant calling, variants were annotated, filtered and prioritized as previously described (Azaiez *et al*, 2015; Booth *et al*, 2015). Briefly, variants were initially filtered based on depth (> 10) and quality score (> 30). Variant was further filtered based on minor allele frequency (MAF) < 2% in Genome Aggregation Database (gnomAD), 1000 Genomes Project database, the Exome Aggregation Consortium (ExAC) and the National Heart, Lung, and Blood Institute (NHLBI) Exome Sequencing Project Exome Variant Server (EVS). Next variants were prioritized based on mutation type (missense, nonsense, indel or splice site), conservation (GERP and PhyloP) and predicted deleteriousness (SIFT, PolyPhen2, MutationTaster, LRT and the Combined Annotation Dependent Depletion (CADD)). Additionally, samples were analysed for CNVs using a sliding window method to assess read-depth ratios (Nord *et al*, 2011). The two reported variants and relevant data have been submitted to the University of Iowa Deafness Variation Database (http://deafness variationdatabase.org/) for expert curation and integration.

## Statistical analysis

Data were analysed using the GraphPad Prizm software (GraphPad Software, Inc, San Diego, USA). Sample sizes were in line with power calculations carried out to detect biologically meaningful differences between mutants and controls for each measure using estimates of variance from previous similar experiments. Statistical significant difference among wild-type and mutant groups was based on the assumption of normal distribution and was tested by the two-tailed unpaired *t*-test with Welch's correction to account for unequal variances. Pearson's chi-squared test was used for the suspension, contact righting and swimming experiments. Statistical significance was set at *P*-values < 0.05.

**Expanded View** for this article is available online.

## Acknowledgements

We thank Elodie Ey for assistance with the tracking software used for open-field exploratory behaviour. This work was supported by the European Union Seventh Framework Programme under the grant agreement HEALTH-F2-2010-242013

## The paper explained

### Problem

Usher syndrome (USH) is the major cause of deaf-blindness in humans. Usher syndrome type I (USH1) is the most severe of the three clinical forms. It is characterized by congenital profound deafness, balance deficits and retinitis pigmentosa. Six USH1 genes have been identified. The roles of the first five USH1 proteins to be identified—myosin VIIa (USH1B), harmonin (USH1C), cadherin-23 (USH1D), protocadherin-15 (USH1F) and sans (USH1G)—in the correct development and functioning of the hair bundle (the structure responsible for sound reception) are well-documented, but the role of CIB2 (USH1J), encoding calcium- and integrin-binding protein 2, remains elusive.

### Results

We found that mice lacking a functional CIB2 protein were profoundly deaf. CIB2 deletion in mice completely disrupted mechanoelectrical transduction activity in the hair cells of the cochlea only. CIB2 was localized to the stereocilia, and its absence caused the mislocalization of whirlin and integrin α8 in cochlear hair bundles. By contrast to other USH1 mutant mice, $CIB2^{-/-}$ mice presented structural abnormalities of the hair bundles only after birth, with stereocilium regression, rapidly followed by hair-cell death. Our molecular and morphofunctional findings in the vestibular organs and retina of $CIB2^{-/-}$ mice provide no evidence for balance and vision dysfunction in the absence of CIB2. We also sought patients with CIB2 mutations, leading to the identification of two new nonsense mutations predicted to produce CIB2-truncated proteins in patients with nonsyndromic hearing loss without signs of retinitis pigmentosa or vestibular dysfunction.

### Impact

Our findings provide evidence that CIB2 is essential for terminal maturation in the auditory hair bundles and for hair-cell survival in the cochlea. In addition to its critical involvement in mechanoelectrical sound transduction, CIB2 probably also functions as a component of a signalling platform that might influence the integrin-mediated link between the extracellular matrix and the intracellular cytoskeleton of the hair bundle. The molecular and morpho-functional defects of the auditory organ are consistent with the profound hearing loss observed in patients with the DFNB48 form of isolated deafness. However, as shown here in both mice and humans, CIB2 function seems to be dispensable for balance and vision, and therefore, caution is required when providing genetic counselling for families with patients carrying CIB2 mutations, as these mutations may not necessarily lead to Usher syndrome.

(TREATRUSH), the European Commission (Hair bundle ERC-2011-ADG_294570), by French state funds managed by the ANR within the Investissements d'Avenir Programme (ANR-15-RHUS-0001), Laboratoires d'excellence "Labex" Lifesenses (ANR-10-LABX-65), (ANR-11-IDEX-0004-02), (ANR-11-BSV5-0011) and grants from la Fondation RETINA-France, the BNP Paribas Foundation, the FAUN Stiftung and the LHW-Stiftung. The work was also supported in part by Medical Research Council (MC_U142684175) to SD. M.B.; Iran National Science Foundation and the grant number 95S47307 to H.N. and NIDCD RO1s: DC003544, DC002842 and DC012049 to RJ.H.S. PP benefitted from funding received from the European Union's Horizon 2020 research and innovation programme under the Marie Sklodowska-Curie grant agreement No. 665807.

## Author contributions

AE-A conceived the study, analysed data and wrote the manuscript with input from all co-authors; VM, PP, and MC performed research and analysed data; ML, AEm and AL performed experiments and collected the vestibular data; AB and AA generated the CIB2 antibody; TD and DO-P assisted and provided critical expertise with animal husbandry; KTB, HA, KK, HN, RJS conceived the genetic study in humans, analysed the data and interpreted the results; JD and SP assisted with the electrophysiological recordings PA performed audiometric tests; MRB, SDMB and CP provided crucial reagents and support.

## Conflict of interest

The authors declare that they have no conflict of interest.

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
