## [Review Process File · EMBO Molecular Medicine]

CIB2, defective in isolated deafness, is key for auditory hair cell mechanotransduction and survival

Vincent Michel, Kevin T Booth, Pranav Patni, Matteo Cortese, Hela Azaiez, Amel Bahloul, Kimia Kahrizi, Ménélik Labbé, Alice Emptoz, Andrea Lelli, Julie Dégardin, Typhaine Dupont, Asadollah Aghaie, Danuta Oficjalska, Serge Picaud, Hossein Najmabadi, Richard J Smith, Michael R Bowl, Stephen D.M. Brown, Paul Avan, Christine Petit, Aziz El-Amraoui

Corresponding author: Aziz EL-AMRAOUI, Institut Pasteur

Review timeline:

Submission date:	29 May 2017
Editorial Decision:	20 July 2017
Revision received:	23 August 2017
Editorial Decision:	19 September 2017
Revision received:	27 September 2017
Accepted:	02 October 2017

Transaction Report:

Editor: Céline Carret

1st Editorial Decision

20 July 2017

Thank you for the submission of your manuscript to EMBO Molecular Medicine, as a back-to-back with the paper from the Smith's lab. We have now heard back from the three referees whom we asked to evaluate your manuscript.

As you will see from the comments below, the three referees are enthusiastic about the study but do have suggestions and recommendations to further improve conclusiveness and clarity. We would like to encourage you to address these in full, and experimentally when needed, as well as rewrite some of the paper to soften the tone and some antagonistic sentences. When particularly prompted about it, the three referees agreed that merging the two papers would provide a strong clinical aspect to the mouse data that would increase the clinical implications, which is particularly important for our scope.

We would welcome the submission of a revised version within three months for further consideration. Please note that EMBO Molecular Medicine strongly supports a single round of revision and that, as acceptance or rejection of the manuscript will depend on another round of review, your responses should be as complete as possible.

I look forward to receiving your revised manuscript.

***** Reviewer's comments *****

Referee #1 (Remarks):

CIB2 is a member of the calcium and integrin binding protein. Mutations in CIB2 have been reported to cause deafness (DFNB48) and deaf-blindness (Usher Syndrome type 1J). In this report, Michel et al. analyze the role played by CIB2 in the inner ear and retina of a mutant mouse model lacking CIB2. The mouse model was generated through embryonic and ubiquitous suppression of exon 4. This work confirms the deafness phenotype associated with CIB2^{-/-} but shows that CIB2 deletion does not alter vestibular and retinal functions.

CIB2 is one of 6 genes that has been associated with USH1J. Usher syndrome is the leading cause of deaf-blindness. Mutations in USH1 genes are associated with congenital deafness, progressive retinal degeneration leading to blindness and vestibular deficits. While CIB2 has been associated with Usher syndrome, this report describes a very thorough study that confirms the role of CIB2 in hearing but demonstrates lack of vestibular and retinal phenotype in absence of CIB2. The results and conclusions are in agreement with a recent report from Zou et al. (2017) that demonstrates that absence of CIB2 does not affect the localization of USH2 protein in hair cells. The manuscript is well written, with figures that include detailed diagrams guiding the readers and summarizing the findings.

Because CIB2 is expressed in many other tissues, this manuscript is of interest to a broader audience and very relevant to the research community focused on the study of genes associated with deafness phenotype.

Major comments:

1- CIB2^{-/-} mice were generated using CIB2 fl/fl mice with two flox sites flanking exon 4 (common to the three splice forms of CIB2). PGK-cre mice were used to obtain CIB2 null mice. PGK is a phosphoglycerate kinase which is expressed ubiquitously in the embryo. The assumption is that the cre line will lead to excision of exon 4 of the CIB2 gene and result in absence of CIB2. Immunostaining of the inner ear sensory epithelium shows absence of CIB2 staining in hair cells of the cochlea. However it remains possible that incomplete cre recombination could lead to partial expression of CIB2. Providing western blots showing total absence of CIB2 in the inner ear, retina and skeletal muscles (which have high CIB2 expression) would further validate the mouse model and alleviate concerns that absence of vestibular and retinal phenotype is not the result of maintained CIB2 expression in these tissues.

2- Related to the question above, the commercial antibody from Abcam labeled both control and mutant mice which lead to the conclusion that this antibody was not specific. The second antibody was generated to target a sequence downstream of exon4 and confirmed absence of protein in cochlear hair cells of mutant mice. While the initial staining may be due to non-specificity, it may also suggest that a truncated CIB2 protein is expressed. Could this truncated protein lead to preservation of function in the vestibular system and the retina? Indeed, while no CIB2 staining is seen in control vestibular hair cells (Fig 2D), CIB2 has been previously shown to be expressed in their stereocilia tip (Riazuddin et al. 2012).

3- The only mutation which has been associated with USH1J, c.192G>A (p.GLU64Asp), is localized in exon 3. This mutation does not lead to absence of CIB2 but instead has been hypothesized to affect integrin binding which in turn may affect downstream signaling pathways. An important question therefore remains regarding whether such mutation in CIB2 may affect balance and vision. For example, while absence of CIB2 might be compensated by CIB1 in some

tissues, the c.192G>A mutation might instead affect CIB2 function (such as interaction with integrin) and prevent compensation by CIB1. These points should be discussed here.

4- SEM shows fairly well preserved IHCs bundles up to P18. While MET current are shown to be absent in neonatal OHCs, it would be interesting to see if MET currents are preserved in IHCs. Such result would suggest that CIB2 is also dispensable for IHCs (assuming the no CIB2 is expressed here).

Minor:

- 1- The title of the manuscript should be revised unless total absence of CIB2 protein expression after PGK-cre excision of exon4 is demonstrated.
- 2- Question marks appear on figure EV3 which presumably do not belong here.
- 3- Figure EV5: "integrin activation"

Referee #2 (Comments on Novelty/Model System):

The mouse is an excellent model for studying inner ear defects and deafness in humans.

Referee #2 (Remarks):

The work describes a mouse mutant representing CIB2 deafness. Previous research demonstrated that pathogenic variants in CIB2 lead to DFNB48 deafness and Usher syndrome IJ, presenting deafness, vestibular defects and blindness. The goal of this work was to dissect the function of CIB2 with respect to deafness and Usher syndrome. An analysis was performed relating to hearing and behavioral defects, with an emphasis on vestibular function. CIB2 interacting proteins were examined in the mutants relative to wild type mice, including myosin VIIa, harmonin, protocadherin-15, whirlin. While the former appeared to have a normal expression pattern and levels in the mutant, whirlin had some abnormal expression at the base of the stereocilia. Integrin and CIB1 expression was changed in the CIB2 mutants, specifically in the cochlea.

The defects in the mice began after birth and there were no apparent abnormalities in the vestibular system or in the retina. Moreover, the defects occurred after birth, which is different than the other known USH-associated genes. As a result, the authors make a strong statement that mutation in CIB2 are not causative for Usher syndrome, despite the previous evidence demonstrating this association in patients, zebrafish and Drosophila.

Comments

Remove from Abstract: "...casting doubts over its causality in Usher syndrome."

Overall, I find the emphasis of the paper - to refute the involvement of CIB2 association with Usher syndrome, to be overdone, as if this is the main finding of the paper. It appears the authors were attempting to emphasize this point, rather than focusing on the scientific data presented - the very fine analysis of CIB2 function in the inner ear. The characterization of the mouse and the analysis of interacting proteins is done very well and thoroughly, and this crucial finding for the inner ear, is what should be emphasized. Moreover, the authors do not outline considerations of why there could be a discrepancy with previous data. The previous data was strong with respect to mutations found in patients, and data from zebrafish and Drosophila. The KO could be functioning differently than the specific mutations found in patients, the genetic background could be a factor, etc. This is not necessarily needed, unless the authors continue to insist on making this point the critical finding of the paper.

Last sentence Introduction:

Altogether, our findings cast doubts over CIB2 causality in Usher syndrome, which would thus impact the genetic counselling in patients with CIB2 mutations.

Change to: Altogether, our findings demonstrate that CIB2 variants may not lead to Usher syndrome, suggesting that caution should be taken when providing genetic counseling for patients with CIB2 mutations.

"These findings suggest that, for the purpose of diagnosis (Idan et al, 2013; Koffler et al, 2015) and genetic counselling in families affected by USH, the classification of CIB2 deficiency as a form of Usher syndrome should be reconsidered, to improve decision-making for affected patients. Change "reconsidered" to "approached with caution"

"...Hispanic families, all of which have been implicated in the DFNB48 form of isolated deafness (refs)."
What is "refs"?

"However, sequencing studies on 427 patients with USH have identified shared mutations in n USH individuals, covering all Usher genes, but no CIB2 mutations were detected in any of these patients (Bonnet et al, 2016)."
What is "n"?

Referee #3 (Remarks):

The manuscript entitled "Profound deafness but no vestibular and retinal defects in mice lacking the CIB2/USH1J protein" by Michel et al deals with phenotype analysis of CIB2^{-/-} mice. CIB2, the calcium- and integrin-binding protein 2, has been previously found as cause for isolated deafness (DFNB48) and human Usher syndrome type-I (USH1J) characterized by congenital profound deafness, balance defects and blindness. In the present study, the authors show that the knock-out of Cib2 in mice causes profound deafness, but no vestibular defects or any retinal phenotype which is different to USH1 in human. In addition, they show, that in comparison to USH1 mutant mice, the structural abnormalities in the cochlear hair cells of CIB2^{-/-} mice much later leading to stereocilia regression and to cell death which is also different to known USH1 models.

The manuscript is well written and illustrated. It provides novel data of high interest in the field of the molecular genetics and cell biology of hearing. Furthermore, it provides reasonable doubts that CIB2 is related to USH type-1.

Nevertheless, have few points of criticism:

- (1) I miss a discussion whether or not CIB2 associates with USH2, which is characterized by progressive hearing loss and no vestibular organ dysfunctions in the inner ear. In addition, the authors point out that CIB2^{-/-} show a phenotype retinal phenotype comparable to USH1 patient. However, they hide that none of the USH1 mouse models develop such a phenotype in the eye. Therefore, the absence of a retinal phenotype does not support the hypothesis that CIB2 is not related to USH1. The same authors have previously also shown that all USH1 proteins are present in calyceal processes, which extend from photoreceptor inner segment to support the outer segment against mechanical stress in non-rodent vertebrate species. An absence from the calyceal processes would support the hypothesis above.
- (2) I encourage the authors to provide immunolocalization of CIB2 in photoreceptors from species with calyceal processes.
- (3) To explain the absence of a dysfunction of the vestibular hair cells in CIB2^{-/-} mice, the authors speculate that other CIB molecules may compensate the lack of CIB2 in vestibular cells. The occurrence of such compensatory mechanisms in the vestibular organs of CIB2^{-/-} mice should be tested.

All in all the present study enlightens the role CIB2 as a linker through integrins, namely integrin $\alpha 8$ to the EMC in cochlear hair cells, but not in the vestibular organs, which arise casting doubts over its causality in Usher syndrome. Later finding is of clinical - diagnostic relevance.

1st Revision - authors' response

23 August 2017

Referee #1 (Remarks):

CIB2 is a member of the calcium and integrin binding protein. Mutations in CIB2 have been reported to cause deafness (DFNB48) and deaf-blindness (Usher Syndrome type 1J). In this report, Michel et al. analyze the role played by CIB2 in the inner ear and retina of a mutant mouse model lacking CIB2. The mouse model was generated through embryonic and ubiquitous suppression of

exon 4. This work confirms the deafness phenotype associated with *CIB2*^{-/-} but shows that *CIB2* deletion does not alter vestibular and retinal functions.

CIB2 is one of 6 genes that has been associated with *USH1J*. Usher syndrome is the leading cause of deaf-blindness. Mutations in *USH1* genes are associated with congenital deafness, progressive retinal degeneration leading to blindness and vestibular deficits. While *CIB2* has been associated with Usher syndrome, this report describes a very thorough study that confirms the role of *CIB2* in hearing but demonstrates lack of vestibular and retinal phenotype in absence of *CIB2*. The results and conclusions are in agreement with a recent report from Zou et al. (2017) that demonstrates that absence of *CIB2* does not affect the localization of *USH2* protein in hair cells. The manuscript is well written, with figures that include detailed diagrams guiding the readers and summarizing the findings.

Because *CIB2* is expressed in many other tissues, this manuscript is of interest to a broader audience and very relevant to the research community focused on the study of genes associated with deafness phenotype.

Major comments:

1- *CIB2*^{-/-} mice were generated using *CIB2* fl/fl mice with two flox sites flanking exon 4 (common to the three splice forms of *CIB2*). PGK-cre mice were used to obtain *CIB2* null mice. PGK is a phosphoglycerate kinase which is expressed ubiquitously in the embryo. The assumption is that the cre line will lead to excision of exon 4 of the *CIB2* gene and result in absence of *CIB2*. Immunostaining of the inner ear sensory epithelium shows absence of *CIB2* staining in hair cells of the cochlea. However it remains possible that incomplete cre recombination could lead to partial expression of *CIB2*. Providing western blots showing total absence of *CIB2* in the inner ear, retina and skeletal muscles (which have high *CIB2* expression) would further validate the mouse model and alleviate concerns that absence of vestibular and retinal phenotype is not the result of maintained *CIB2* expression in these tissues.

We have already successfully used pGK-Cre mice to produce total knockout mice for many Usher proteins, including *Ush1c* (Lefèvre et al, 2008; Michalski et al, 2009) and *Ush1g* (Caberlotto et al, 2011). All these previously generated *Ush1* mutant mice display congenital profound deafness and balance deficits, attesting to the efficient expression of the Cre recombinase in both sense organs, and faithfully mimicking the dual sensory deficit of the inner ear observed in *USH1* patients. The lack of vestibular dysfunction in our *CIB2*^{-/-} mice is, therefore, unlikely to be due to an incomplete penetrance of the Cre gene under the control of the pGK promoter. Indeed, like our *CIB2*^{-/-} mice (this study), *CIB2*^{tm1a/tm1a} (total knockout mice obtained by crossing with b-actin (*ACTB*)-Cre mice) and *CIB2*^{F91S/F91S} (constitutive knock-in mouse, reproducing the human p.F91S *CIB2* mutation) mice (Giese et al. 2017) display profound hearing defects with no vestibular dysfunction. Last, but not least, RT-PCR on several tissues and organs from wild-type and *CIB2*^{-/-} mice revealed no evidence of *CIB2* transcripts containing exon 4, confirming the deletion of this exon in all tested tissues (inner ear, eye, brain, muscle, kidney and testis) (see Fig. 1A, and B, and corresponding legend (page 28, lines 5-7; a copy of the figure is shown below). Transcripts lacking exon 4 and harbouring a premature termination codon were detected; these transcripts were predicted to encode a truncated *CIB2* protein containing the N-terminal region (amino acids 1-65 (+ 12 unrelated aa before the stop codon)), but lacking the C-terminal region (aa 65-187) with its three EF domains (see Fig. 1A, lower panel).

Together, these findings indicate that a total loss of *CIB2* function is detrimental only to hearing. The presence of *CIB2* nonsense mutations in patients with profound deafness with no signs of retinitis pigmentosa or vestibular dysfunction (Riazuddin et al, 2012; Seco et al, 2016) and this study (see text, pages 9 10, and new figure 9) provides further support for our conclusions.

2- Related to the question above, the commercial antibody from Abcam labeled both control and mutant mice which lead to the conclusion that this antibody was not specific. The second antibody was generated to target a sequence downstream of exon4 and confirmed absence of protein in cochlear hair cells of mutant mice. While the initial staining may be due to non-specificity, it may also suggest that a truncated CIB2 protein is expressed. Could this truncated protein lead to preservation of function in the vestibular system and the retina? Indeed, while no CIB2 staining is seen in control vestibular hair cells (Fig 2D), CIB2 has been previously shown to be expressed in their stereocilia tip (Riazuddin et al. 2012).

As mentioned above, even if produced, the mutated truncated CIB2 protein would consist of the N-terminal region only (**amino acids 1-65; see Fig. 1A (copy above)**). Modelling predictions indicate that the truncated form of CIB2 is non-functional, as it lacks the C-terminal region (aa 66-187), which contains the three EF domains and the C-terminal PDZ-binding motif (**see Fig. 1A**). The findings in human deaf patients harbouring stop mutations in the N-terminal region of CIB2 (Seco et al. 2016, and this study (**see Fig. 9A,B**)) clearly indicate that the N-terminal fragment alone is insufficient to maintain normal CIB2 function.

In the conditions used here, we were unable to detect specific *CIB2* expression with the commercial Abcam antibody. In such conditions, and when knockout mice are available for a given protein, we make use only of immunostaining that differs between wild-type and knockout mice as obtained by our homemade anti-CIB2 antibodies and shown in figure 2. By contrast, under our used conditions, the commercial Abcam antibody (directed against a human internal CIB2 sequence) does not differentiate between wild-type and knockout mice, despite having been shown to label vestibular stereocilia tips ((Riazuddin et al, 2012), Fig. 2D; in 2012 no CIB2 mice was available). No information is available concerning the internal sequence against which the Abcam antibody (<http://www.abcam.com/cib2-antibody-ab111908.html>) is directed, making it difficult to determine whether the labelling corresponds to the truncated CIB2 or to cross-reactions with unrelated proteins.

3- The only mutation which has been associated with USH1J, c.192G>A (p.GLU64Asp), is localized in exon3. This mutation does not lead to absence of CIB2 but instead has been hypothesized to affect integrin binding which in turn may affect downstream signaling pathways. An important question therefore remains regarding whether such mutation in CIB2 may affect balance and vision. For example, while absence of CIB2 might be compensated by CIB1 in some tissues, the c.192G>A mutation might instead affect CIB2 function (such as interaction with integrin) and prevent compensation by CIB1. These points should be discussed here.

We have rewritten this part of the manuscript, to introduce genetic and clinical data for *CIB2* mutations in humans, and information about the potential impact of the mutations on protein production, distribution and function (**see page 14, lines 18-33**). Following on from the initial identification of *CIB2* mutations in Pakistani and Turkish families (Riazuddin et al, 2012), new *CIB2* mutations have been found in Dutch, Caribbean Hispanic (Patel et al, 2015; Seco et al, 2016), Iranian and Palestinian Arab (**this study, see Fig. 9A,B**) families. Of the nine mutations (e.g. nonsense, missense, splicing) reported to date, only the p.Glu64Asp missense variant, described in one Pakistani family, has been shown to cause USH1J (Riazuddin et al, 2012)(**see diagram in Fig. 1A**). Two other missense mutations, p.Arg66Trp located 2 amino acids downstream from the USH1J variant and p.Phe91Ser located in the first EF domain, have also been reported to affect the interaction of CIB2 with integrin, without affecting CIB2 protein levels or targeting to the stereocilia (Patel et al, 2015; Riazuddin et al, 2012; Seco et al, 2016). Nevertheless, all known patients with either the p.Arg66Trp or the p.Phe91Ser mutation have the DFNB48 form of isolated deafness, with normal vestibular and ocular functions (Riazuddin et al, 2012)(Patel et al, 2015; Seco et al, 2016). Thus, overall, the available data suggest that mutations affecting either calcium binding, or interaction with integrins, result in profound deafness with no signs of retinitis pigmentosa or vestibular dysfunction (**see text page 14, lines 18-33**). Additional data are required to determine whether and how defects of this gene, which appears to be the only exception to the pattern for USH1 genes observed to date, can cause hearing, vestibular, and ocular defects.

4- SEM shows fairly well preserved IHCs bundles up to P18. While MET current are shown to be absent in neonatal OHCs, it would be interesting to see if MET currents are preserved in IHCs. Such result would suggest that CIB2 is also dispensable for IHCs (assuming the no CIB2 is expressed here).

In the revised manuscript, we cite the recent work by **Giese *et al.* (2017)** showing that, like the auditory outer hair cells (OHCs) (this study), the inner hair cells (IHCs) also display a total absence of mechano-electrical transduction (MET) activity on P6. We cite this paper here to extend our conclusions about the critical role of CIB2 in MET current responses to all hair cells (**see page 11, lines 19-22**), consistent with the profound hearing loss (thresholds over 90-100 dB SPL for all sound frequencies) observed in *CIB2*^{-/-} mice (**Fig. 1B**) and patients with *CIB2* mutations (**Fig. 9B**).

Minor:

1- *The title of the manuscript should be revised unless total absence of CIB2 protein expression after PGK-cre excision of exon4 is demonstrated.*

The total deletion of exon 4 after pGK-Cre excision was confirmed in the studied tissues (inner ear and eye), and in other tissues (brain, muscle, kidney and testis (**see Fig. 1B, copy above at page 3**)). We have nonetheless modified the title in accordance with the key function of this protein in hearing (**see page 1**).

2- *Question marks appear on figure EV3 which presumably do not belong here.*

The question marks have been deleted.

2- *Figure EV5: "integrin activation"*

This part of the figure (text content and positioning) has been modified accordingly. The **new version of the figure** also illustrate the recent finding reported by Giese *et al.* (Giese *et al.*, 2017) of direct interaction between CIB2 and the TMC1/TMC2 channel complex (**see discussion, page 11, lines 19-22**).

Referee #2 (Remarks):

The work describes a mouse mutant representing CIB2 deafness. Previous research demonstrated that pathogenic variants in CIB2 lead to DFNB48 deafness and Usher syndrome IJ, presenting deafness, vestibular defects and blindness. The goal of this work was to dissect the function of CIB2 with respect to deafness and Usher syndrome. An analysis was performed relating to hearing and behavioral defects, with an emphasis on vestibular function. CIB2 interacting proteins were examined in the mutants relative to wild type mice, including myosin VIIa, harmonin, protocadherin-15, whirlin. While the former appeared to have a normal expression pattern and levels in the mutant, whirlin had some abnormal expression at the base of the stereocilia. Integrin and CIB1 expression was changed in the CIB2 mutants, specifically in the cochlea.

The defects in the mice began after birth and there were no apparent abnormalities in the vestibular system or in the retina. Moreover, the defects occurred after birth, which is different than the other known USH-associated genes. As a result, the authors make a strong statement that mutation in CIB2 are not causative for Usher syndrome, despite the previous evidence demonstrating this association in patients, zebrafish and Drosophila.

Comments

Remove from Abstract: "...casting doubts over its causality in Usher syndrome."

This part of the abstract has been deleted.

Overall, I find the emphasis of the paper - to refute the involvement of CIB2 association with Usher syndrome, to be overdone, as if this is the main finding of the paper. It appears the authors were attempting to emphasize this point, rather than focusing on the scientific data presented - the very fine analysis of CIB2 function in the inner ear. The characterization of the mouse and the analysis of interacting proteins is done very well and thoroughly, and this crucial finding for the inner ear, is what should be emphasized. Moreover, the authors do not outline considerations of why there could be a discrepancy with previous data. The previous data was strong with respect to mutations found in patients, and data from zebrafish and Drosophila. The KO could be functioning differently than the specific mutations found in patients, the genetic background could be a factor, etc. This is not

necessarily needed, unless the authors continue to insist on making this point the critical finding of the paper.

We agree with the reviewer that the fine analysis of CIB2 function through detailed morpho-physiological characterization of the inner ear in *CIB2*^{-/-} mice is, and should be, a key message of the manuscript. The comparison of the phenotype of the mutant mice with that of other USH1 mice is also a key element. We have rewritten the corresponding parts of the manuscript, to soften the tone and keep the message focused on the data and their potential impact (text in blue in the introduction, and discussion).

Last sentence Introduction:

Altogether, our findings cast doubts over CIB2 causality in Usher syndrome, which would thus impact the genetic counselling in patients with CIB2 mutations.

Change to: Altogether, our findings demonstrate that CIB2 variants may not lead to Usher syndrome, suggesting that caution should be taken when providing genetic counseling for patients with CIB2 mutations.

This part of the text has been modified accordingly (see page 4, lines 15-17).

"These findings suggest that, for the purpose of diagnosis (Idan et al, 2013; Koffler et al, 2015) and genetic counselling in families affected by USH, the classification of CIB2 deficiency as a form of Usher syndrome should be reconsidered, to improve decision-making for affected patients. Change "reconsidered" to "approached with caution"

This part of the text has been modified accordingly (see page 14, last sentence).

"...Hispanic families, all of which have been implicated in the DFNB48 form of isolated deafness (refs)."

What is "refs"?

"However, sequencing studies on 427 patients with USH have identified shared mutations in n USH individuals, covering all Usher genes, but no CIB2 mutations were detected in any of these patients (Bonnet et al, 2016)." What is "n"?

n = 421 : This part of the text has been modified accordingly (see page 14, last sentence).

The references covering the identified *CIB2* mutations in patients are cited and include the number of patients with identified mutations in USH genes (421 of 427; (Bonnet et al, 2016)).

Referee #3 (Remarks):

The manuscript entitled "Profound deafness but no vestibular and retinal defects in mice lacking the CIB2/USH1J protein" by Michel et al deals with phenotype analysis of CIB2^{-/-} mice. CIB2, the calcium- and integrin-binding protein 2, has been previously found as cause for isolated deafness (DFNB48) and human Usher syndrome type-I (USH1J) characterized by congenital profound deafness, balance defects and blindness. In the present study, the authors show that the knock-out of Cib2 in mice causes profound deafness, but no vestibular defects or any retinal phenotype which is different to USH1 in human. In addition, they show, that in comparison to USH1 mutant mice, the structural abnormalities in the cochlear hair cells of CIB2^{-/-} mice much later leading to stereocilia regression and to cell death which is also different to known USH1 models.

The manuscript is well written and illustrated. It provides novel data of high interest in the field of the molecular genetics and cell biology of hearing. Furthermore, it provides reasonable doubts that CIB2 is related to USH type-I.

We thank the reviewer for his/her positive comments on our work.

Nevertheless, have few points of criticism:

(1) I miss a discussion whether or not CIB2 associates with USH2, which is characterized by progressive hearing loss and no vestibular organ dysfunctions in the inner ear.

In addition, the authors point out that CIB2^{-/-} show a phenotype retinal phenotype comparable to USH1 patient. However, they hide that none of the USH1 mouse models develop such a phenotype

in the eye. Therefore, the absence of a retinal phenotype does not support the hypothesis that *CIB2* is not related to *USH1*. The same authors have previously also shown that all *USH1* proteins are present in calyceal processes, which extend from photoreceptor inner segment to support the outer segment against mechanical stress in non-rodent vertebrate species. An absence from the calyceal processes would support the hypothesis above.

(2) I encourage the authors to provide immunolocalization of *CIB2* in photoreceptors from species with calyceal processes.

In the revised manuscript, we include our observations regarding *CIB2* expression in the retina, which provide no support for a potential link between *CIB2* and Usher syndrome (see page 14, lines 13-18, and Appendix Fig. S1 & legend: a copy of the figure is shown page 9). We found no unambiguous difference in *CIB2* immunolabeling between wild-type and *CIB2*^{-/-} mice, irrespective of the anti-*CIB2* antibody used (see supplementary Figure 1, appendix), ruling out the conclusion that this staining was specific to *CIB2*. The experiments were repeated several times, but we found no *CIB2*-specific immunostaining in the calyceal processes of macaque photoreceptors (such as that observed for *USH1* proteins, (Sahly et al, 2012)) or in the periciliary ridge of mouse photoreceptors (such as that observed for *USH2* proteins, (Sahly et al, 2012)). Consistent with the reviewer's statement (underlined above), our studies of expression in the retina provided no evidence of a potential link between *CIB2* and Usher-associated substructures.

Supplementary Figure S1, Appendix: *CIB2* immunostaining in the retina

*(3) To explain the absence of a dysfunction of the vestibular hair cells in *CIB2*^{-/-} mice, the authors speculate that other *CIB* molecules may compensate the lack of *CIB2* in vestibular cells. The occurrence of such compensatory mechanisms in the vestibular organs of *CIB2*^{-/-} mice should be tested.*

All in all the present study enlightens the role CIB2 as a linker through integrins, namely integrin $\alpha 8$ to the EMC in cochlear hair cells, but not in the vestibular organs, which arise casting doubts over its causality in Usher syndrome. Later finding is of clinical - diagnostic relevance.

We have rewritten this part of the manuscript to include recent information from Giese *et al.* (Giese *et al.*, 2017) concerning *CIB* transcript (*CIB 1 to 4*) levels in the inner ear (see discussion, page 13, lines 25-36). RT-PCR analyses at P12 showed that *CIB3* levels in the vestibular sensory epithelia were almost eight times higher than those of the cochlea (Giese *et al.*, 2017), indicating that *CIB3* may play a crucial role in vestibular function. In the absence of a functional *CIB2*, significant upregulation was observed only for *CIB1*, specifically in the cochlea (Giese *et al.*, 2017). Together with our findings for *CIB2* and *CIB1*, these data suggest that *CIB1* and *CIB2* play complementary key roles in the cochlea, and that the vestibular organs are relatively insensitive to *CIB2* dysfunction (see discussion, page 13, lines 25-36). Further studies are required to determine the precise role of *CIB* paralogs in the different inner ear compartments, including, in particular, whether a lack of *CIB3* is detrimental to vestibular function.

REFERENCES:

Bonnet C, Riahi Z, Chantot-Bastaraud S, Smaghe L, Letexier M, Marcaillou C, Lefevre GM, Hardelin JP, El-Amraoui A, Singh-Estivalet A, Mohand-Said S, Kohl S, Kurtenbach A, Sliesoraityte I, Zobor D, Gherbi S, Testa F, Simonelli F, Banfi S, Fakin A, Glavac D, Jarc-Vidmar M, Zupan A, Battelino S, Martorell Sampol L, Claveria MA, Catala Mora J, Dad S, Moller LB, Rodriguez Jorge J, Hawlina M, Auricchio A, Sahel JA, Marlin S, Zrenner E, Audo I, Petit C (2016) An innovative strategy for the molecular diagnosis of Usher syndrome identifies causal biallelic mutations in 93% of European patients. *Eur J Hum Genet* **24**: 1730-1738

Caberlotto E, Michel V, Foucher I, Bahloul A, Goodyear RJ, Pepermans E, Michalski N, Perfettini I, Alegria-Prevot O, Chardenoux S, Do Cruzeiro M, Hardelin JP, Richardson GP, Avan P, Weil D, Petit C (2011) Usher type 1G protein sans is a critical component of the tip-link complex, a structure controlling actin polymerization in stereocilia. *Proc Natl Acad Sci USA* **108**: 5825-5830

Giese APJ, Tang YQ, Sinha GP, Bowl MR, Goldring AC, Parker A, Freeman MJ, Brown SDM, Riazuddin S, Fettiplace R, Schafer WR, Frolenkov GI, Ahmed ZM (2017) *CIB2* interacts with *TMC1* and *TMC2* and is essential for mechanotransduction in auditory hair cells. *Nat Commun* **8**: 43

Lefèvre G, Michel V, Weil D, Lepelletier L, Bizard E, Wolfrum U, Hardelin JP, Petit C (2008) A core cochlear phenotype in *USH1* mouse mutants implicates fibrous links of the hair bundle in its cohesion, orientation and differential growth. *Development* **135**: 1427-1437

Michalski N, Michel V, Caberlotto E, Lefèvre GM, van Aken AFJ, Tinevez J-Y, Bizard E, Houbbron C, Weil D, Hardelin J-P, Richardson GP, Kros C, Martin P, Petit C (2009) Harmonin-b, an actin-binding scaffold protein, is involved in the adaptation of mechanoelectrical transduction by sensory hair cells. *Pflügers Arch* **459**: 115-130

Patel K, Giese AP, Grossheim JM, Hegde RS, Delio M, Samanich J, Riazuddin S, Frolenkov GI, Cai J, Ahmed ZM, Morrow BE (2015) A Novel C-Terminal *CIB2* (Calcium and Integrin Binding Protein 2) Mutation Associated with Non-Syndromic Hearing Loss in a Hispanic Family. *PLoS One* **10**: e0133082

Riazuddin S, Belyantseva IA, Giese AP, Lee K, Indzhukulian AA, Nandamuri SP, Yousaf R, Sinha GP, Lee S, Terrell D, Hegde RS, Ali RA, Anwar S, Andrade-Elizondo PB, Sirmaci A, Parise LV, Basit S, Wali A, Ayub M, Ansar M, Ahmad W, Khan SN, Akram J, Tekin M, Cook T, Buschbeck EK, Frolenkov GI, Leal SM, Friedman TB, Ahmed ZM (2012) Alterations of the *CIB2* calcium- and integrin-binding protein cause Usher syndrome type 1J and nonsyndromic deafness DFNB48. *Nat Genet*

Sahly I, Dufour E, Schietroma C, Michel V, Bahloul A, Perfettini I, Pepermans E, Estivalet A, Carette D, Aghaie A, Ebermann I, Lelli A, Iribarne M, Hardelin JP, Weil D, Sahel JA, El-Amraoui

A, Petit C (2012) Localization of Usher 1 proteins to the photoreceptor calyceal processes, which are absent from mice. *J Cell Biol* **199**: 381-399

Seco CZ, Giese AP, Shafique S, Schraders M, Oonk AM, Grossheim M, Oostrik J, Strom T, Hegde R, van Wijk E, Frolenkov GI, Azam M, Yntema HG, Free RH, Riazuddin S, Verheij JB, Admiraal RJ, Qamar R, Ahmed ZM, Kremer H (2016) Novel and recurrent CIB2 variants, associated with nonsyndromic deafness, do not affect calcium buffering and localization in hair cells. *Eur J Hum Genet* **24**: 542-549

2nd Editorial Decision

19 September 2017

Thank you for the submission of your revised manuscript to EMBO Molecular Medicine. We have now received the enclosed reports from the referees who were asked to re-assess it. As you will see the reviewers are now globally supportive and I am pleased to inform you that we will be able to accept your manuscript pending the following final amendments:

- 1) please address the last set of comments-I will not ask for more experiments at this point unless you already have Western Blots to satisfy referee 3. However, carefully go through your text, check for typos and errors; please also pay attention to ref2 and 3's comments regarding careful assumptions but still inclusion of the human data in abstract at least to set your study apart from Giese et al.
- 2) Please carefully check the authors guidelines for formatting your supplemental information: Expanded view and Appendix (see: <http://embomolmed.embopress.org/authorguide#expandedview>)
- 3) Data deposition: you must provide an accession number for the sequencing data before publication.

Please submit your revised manuscript within two weeks. I look forward to seeing a revised form of your manuscript as soon as possible.

***** Reviewer's comments *****

Referee #1 (Remarks for Author):

The revised combined manuscript is much improved. The authors have appropriately addressed the reviewers concerns and have adjusted their publication in view of the recent paper from Giese et al. 2017.

Minor: A few typos and missing or added parenthesis appear in the manuscript. Please check.

Referee #2 (Comments on Novelty/Model System for Author):

The mouse is the ideal model for studying the auditory system, so the model system is more than adequate.

Referee #2 (Remarks for Author):

The last sentence of the abstract should be modified, as it is too speculative. End with "...cochlea." Should not name proteins that are "unidentified" and "probably" and "assuming" is too uncertain for the final text of an abstract

Referee #3 (Remarks for Author):

The authors followed reviewers' suggestions and merged the two submitted manuscript EMM-2017-08094 and EMM-2017-08087 to one EMM-2017-08087-V2 and changed the title to "CIB2, defective in isolated deafness, is key for auditory hair cell mechanotransduction and survival." The

merged manuscript certainly improved and would nicely fit into EMM. Unfortunately, there is some overlap with a publication by Giese et al. 2017 which was recently published during the reviewing process of the present manuscript. In both studies, Cib2 deficient mice were generated and studied revealing the identical phenotypes in hearing which is also indicated in the titles of both stories: "CIB2 ... is essential for mechanotransduction in auditory hair cells" and "CIB2 ... is key for auditory hair cell mechanotransduction ...". In contrast to the published paper, the present manuscript additionally deals with the aspect of the reasonable doubts of CIB2 as a type-1 gene which should be pronounced in the title.

Nevertheless, I would recommend minor revision:

In the supplementary figure S1B, the authors show that CIB2 is not localized in the calyceal processes nor in the in the periciliary ridge region of non-human primate photoreceptor cells where USH1 and USH2 proteins are found.

In figure S1A, the authors show the absence of anti-Cib2 immunofluorescence (besides some background in the outer plexiform layer) in the wild type mouse retinas applying their newly generated polyclonal antibody against CIB2. This finding is in contrast to the findings by Riazuddin and coworkers 2012 on the immunostaining of the mouse retina indicating the expression of Cib2 in several layers of the neuronal retina of the mouse, including the photoreceptor layer. In the latter work a commercial antibody against CIB2 from Abnova was used which was recently validated in Giese et al. (2017) on knock-out tissue. Interestingly, the authors demonstrate in Fig. 1B Cib2 mRNA expression in the eye by RT-PCR. Therefore, I have to ask, where is Cib2 expressed in the eye? The authors should comment on this.

In my opinion comparative Western blot analyses of Cib2 ko and wt retinal tissue applying their newly generated polyclonal antibody against CIB2 are also lacking. Referee 1 already asked for Western blots in a different context.

Further comments:

The legend of figure S1 and the text on page 14 does not correspond. In the legend S1B the authors describe the immunofluorescence staining for only macaque photoreceptor cells, but in the text they talk about a staining in the periciliary ridge of mouse photoreceptor cells.

In figure S1 "C" should be changed to "B".

As I mentioned in my previous review, the manuscript is well written and illustrated. It includes novel data of high interest in the field of the molecular genetics. Furthermore, it provides reasonable doubts that CIB2 is related to USH type-1 important for clinical diagnostic strategies.

2nd Revision - authors' response

27 September 2017

Referee #1 (Remarks for Author):

The revised combined manuscript is much improved. The authors have appropriately addressed the reviewers concerns and have adjusted their publication in view of the recent paper from Giese et al. 2017.

Minor: A few typos and missing or added parenthesis appear in the manuscript. Please check. This has been changed, accordingly.

Referee #2 (Comments on Novelty/Model System for Author):

The mouse is the ideal model for studying the auditory system, so the model system is more than adequate.

Referee #2 (Remarks for Author):

The last sentence of the abstract should be modified, as it is too speculative. End with "...cochlea." Should not name proteins that are "unidentified" and "probably" and "assuming" is too uncertain for the final text of an abstract

This has been changed, accordingly.

Referee #3 (Remarks for Author):

The authors followed reviewers' suggestions and merged the two submitted manuscript EMM-2017-08094 and EMM-2017-08087 to one EMM-2017-08087-V2 and changed the title to "CIB2, defective in isolated deafness, is key for auditory hair cell mechanotransduction and survival." The merged manuscript certainly improved and would nicely fit into EMM. Unfortunately, there is some overlap with a publication by Giese et al. 2017 which was recently published during the reviewing process of the present manuscript. In both studies, *Cib2* deficient mice were generated and studied revealing the identical phenotypes in hearing which is also indicated in the titles of both stories: "CIB2 ... is essential for mechanotransduction in auditory hair cells" and "CIB2 ... is key for auditory hair cell mechanotransduction ...". In contrast to the published paper, the present manuscript additionally deals with the aspect of the reasonable doubts of CIB2 as a type-1 gene which should be pronounced in the title.

Nevertheless, I would recommend minor revision:

In the supplementary figure S1B, the authors show that CIB2 is not localized in the calyceal processes nor in the in the periciliary ridge region of non-human primate photoreceptor cells where USH1 and USH2 proteins are found. In figure S1A, the authors show the absence of anti-Cib2 immunofluorescence (besides some background in the outer plexiform layer) in the wild type mouse retinas applying their newly generated polyclonal antibody against CIB2. This finding is in contrast to the findings by Riazuddin and coworkers 2012 on the immunostaining of the mouse retina indicating the expression of *Cib2* in several layers of the neuronal retina of the mouse, including the photoreceptor layer. In the latter work a commercial antibody against CIB2 from Abnova was used which was recently validated in Giese et al. (2017) on knock-out tissue. Interestingly, the authors demonstrate in Fig. 1B *Cib2* mRNA expression in the eye by RT-PCR. Therefore, I have to ask, where is *Cib2* expressed in the eye? The authors should comment on this.

In my opinion comparative Western blot analyses of *Cib2* ko and wt retinal tissue applying their newly generated polyclonal antibody against CIB2 are also lacking. Referee 1 already asked for Western blots in a different context.

RT-PCR data are more sensitive, and in our study they represent expression of CIB2 in the whole eye. We used these data (Figure 1B) to illustrate the deletion of exon 4 in the *CIB2*^{-/-} mice. In our immunostaining data, we focused on the photoreceptor layer where all other Usher proteins are co-expressed. Under the conditions we used, and at least using our antibody (*see comment below for the commercial antibody against CIB2 from Abnova*), comparisons between wild-type and *CIB2*^{-/-} mice didn't provide evidence of possible expression in these cells. Thus, we have no evidence to assign a given distribution pattern to CIB2 in the photoreceptor cells. Absent or lower amount of protein, inaccessible immunogenic epitopes are additional explanation of absence of protein detection by an antibody. Finally, regarding CIB2 expression in the retina published in 2012 [*text from authors was edited as per their discretion*] The fuzzy "reddish" signal can be observed with most primary antibodies tested on retina cryosections. Some intense immunostaining [*text from authors was edited as per their discretion*] is observed in vessels/capillaries (considered unspecific) and in the outer region of the eye that includes, mostly, the choroid region (surrounding the eye), and overlaps partly with the retinal pigment epithelium [*text from authors was edited as per their discretion*].

Additional studies in both wild-type and *CIB2*^{-/-} mouse samples, using the *abnova* commercial antibody and other home-made anti-CIB2 antibodies, are necessary to confirm or infirm the presence of CIB2 in photoreceptor cells, or in other retinal substructures.

Further comments:

The legend of figure S1 and the text on page 14 does not correspond. In the legend S1B the authors describe the immunofluorescence staining for only macaque photoreceptor cells, but in the text they talk about a staining in the periciliary ridge of mouse photoreceptor cells. In figure S1 "C" should be changed to "B".

The appendix Figure S1 includes data in macaque (A) and in mouse (B): the calyceal processes are present only in macaque, whilst the periciliary ridge region is present in photoreceptors of the two species. The figure (Appendix Figure S1), and the related text (page 14, line 16) have been modified accordingly.

As I mentioned in my previous review, the manuscript is well written and illustrated. It includes novel data of high interest in the field of the molecular genetics. Furthermore, it provides reasonable doubts that CIB2 is related to USH type-1 important for clinical diagnostic strategies.

Corresponding Author Name: EL-AMRAOUI Aziz

Manuscript Number: EMM-2017-08087